# Sparse MoE as a New Treatment: Addressing Forgetting, Fitting, Learning Issues in Multi-Modal Multi-Task Learning

## Abstract

Sparse Mixture-of-Experts (SMoE) is a promising paradigm that can be easily tailored for multi-task learning. Its conditional computing nature allows us to organically allocate relevant parts of a model for performant and efficient predictions. However, several under-explored pain points persist, especially when considering scenarios with both multiple modalities and tasks: ① *Modality Forgetting Issue.* Diverse modalities may prefer conflicting optimization directions, resulting in ineffective learning or knowledge forgetting; ② *Modality Fitting Issue.* Current SMoE pipelines select a fixed number of experts for all modalities, which can end up over-fitting to simpler modalities or under-fitting complex modalities; ③ *Heterogeneous Learning Pace.* The varied modality attributes, task resources (*i.e.,* the number of input samples), and task objectives usually lead to distinct optimization difficulties and convergence. Given these issues, there is a clear need for a systematic approach to harmonizing multi-model and multi-task objectives when using SMoE. We aim to address these pain points, and propose a new Sparse MoE framework for Multi-Modal Multi-task learning, *a.k.a.*, SM$^4$, which (1) disentangles model spaces for different modalities to mitigate their optimization conflicts; (2) automatically determines the modality-specific model size (*i.e.*, the number of experts) to improve fitting; and (3) synchronizes the learning paces of disparate modalities and tasks based on training dynamics in SMoE like the entropy of routing decisions. Comprehensive experiments validate the effectiveness of SM$^4$, which outperforms previous state-of-the-art across 3 task groups and 11 different modalities with a clear performance margin (*e.g.*, $\geq 1.37\%$) and a substantial computation reduction ($46.49\% \sim 98.62\%$). Code is included in the supplement.

## 1 Introduction

Multi-modal multi-task learning (*a.k.a.*, M$^3$TL) aims to resolve different objectives simultaneously. Each objective takes various modalities as input, which is a common scenario required in real-world applications like robotics (Sun et al., 2022) and auto-driving systems (Lee, 2021). Many prior works have extended unimodal transformers (Vaswani et al., 2017) to work on multiple multi-modal tasks (Su et al., 2020; Cho et al., 2021; Hu & Singh, 2021; Lu et al., 2019; Akbari et al., 2021). In their ideal setup, the information from different modalities and tasks prompts each other for better performance. However, the optimization complexity of this sophisticated system limits the development of effective solutions (Sener & Koltun, 2018; Peng et al., 2022). Recently, the sparsely-gated Mixture-of-Experts (SMoE) method was identified as a powerful tool for these complex training dynamics of multi-task (Fan et al., 2022; Zhou et al., 2022a; Gupta et al., 2022; Hazimeh et al., 2021a; Ma et al., 2018a; Chen et al., 2023b) or multi-modal (Shen et al., 2023b;a; Sun et al., 2023) learning. SMoE selects a subset of experts for a task or modality per input sample, and has led to state-of-the-art performance (Mustafa et al., 2022; Chen et al., 2023a).

Despite preliminary success in M$^3$TL, when we try to model multiple modalities and multiple tasks through a *single* network (*e.g.*, SMoE), several under-explored pain points persist: ① *Modality Forgetting Issue.* Considering a model trained on multiple modalities, diverse modalities can prefer conflicting optimization directions within shared parameters. For instance, recent works have shown that there are negative cosine similarities between gradients from different modalities (Alamri et al.,

2019; Javaloy et al., 2022; Chen et al., 2020; Wang et al., 2020a). Such gradient disagreement within a network can lead to inferior learning, or, in the worst case, the multi-modal model can degenerate into a "single-modal" model that only learns the modality with dominant gradients (Peng et al., 2022). ② *Modality Fitting Issue.* The vanilla SMoE architecture activates a *fixed* number of experts to deal with each input. However, some modalities are easier to learn than others. Using too many experts for a simple modality may cause overfitting, while too few experts for complex modalities may cause underfitting (Wang et al., 2020a). As more modalities are introduced, this weakness likely grows. ③ *Heterogeneous Learning Pace.* Current SMoE solutions also have yet to adapt to different objectives between tasks. In reality, the objectives can vary substantially. Consider writing robots, for example. A writing robot must handle two tasks: *object pose prediction* and *digit number classification*. Pose prediction uses images, force sensors, proprioception sensors, and robotic control signals as observations to predict the object's position after the robot executes the control signal. Digit classification uses images and audio to output the corresponding number. Each objective differs significantly in terms of modality attributes, task resources, and task objectives, which leads to great heterogeneity in their optimization pace or convergence (Zhang & Yeung, 2011; Sun et al., 2020; Kollias et al., 2021).

In this paper, we incorporate innovative designs to upgrade the original SMoE algorithm for Multi-Modal Multi-task learning, herein termed $\texttt{SM}^4$, tackling the aforementioned barriers. Specifically, $\texttt{SM}^4$ facilitates learning from three perspectives: ① (*Model*) $\texttt{SM}^4$ customizes the SMoE layer into both the feed-forward networks (FFN) and multi-head self-attention modules (MSA) in transformers, which sufficiently disentangles network parameter space for different modalities and tasks. As shown in Figure 2, the gradient conflict is then greatly reduced. ② (*Routing*) An adaptive expert allocation mechanism is proposed to automatically determine the number of selected experts (or model capacity) for different modalities. $\texttt{SM}^4$ monitors the modality-specific training dynamics (*e.g.*, validation loss), which serve as a reliable indicator to activate more or less experts to mitigate possible under-fitting or over-fitting, respectively. Figure 2 shows an example of how $\texttt{SM}^4$ mitigates overfitting in a simple modality. ③ (*Optimization*) For each modality in one task, $\texttt{SM}^4$ adopts adaptive learning paces based on the convergence status of modality-specific routing policies to synchronize the optimization of multiple objectives. Our contributions can be summarized as follows:

* ⋆ We propose $\texttt{SM}^4$, a framework for multi-modal multi-task learning, which contains tailored SMoE layers for replacing FFN and sparse mixture-of-attention layers as the alternative for vanilla MSA modules in transformers. This disentangles network parameters and alleviates gradient conflicts between different modalities and tasks.

* ⋆ We identify two essential factors in M$^3$TL, *i.e., modality fitting issue* and *heterogeneous learning pace*, which are unstudied by existing SMoE approaches. We then propose corresponding *adaptive expert allocation* and *adaptive learning paces*.

* ⋆ Extensive empirical investigations over 3 representative task groups and 11 diverse modalities consistently validate the effectiveness of $\texttt{SM}^4$. Our method surpasses dense models with similar computational costs, and shows substantial performance improvements; $\texttt{SM}^4$ outperforms existing M$^3$TL SOTA using only 1.38% to 53.51% of their computational cost.

## 2 RELATED WORK

**Multi-modal and Multi-task Learning.** There has been a long history of work on multi-modal learning (Makadia et al., 2008; Weston et al., 2011; Frome et al., 2013; Socher et al., 2013; Antol et al., 2015; Goyal et al., 2017; Ramesh et al., 2022; Saharia et al., 2022; Agrawal et al., 2017; Yang et al., 2016; Dai et al., 2022; Jaegle et al., 2021; 2022) and multi-task learning (Xue et al., 2007; Strezoski et al., 2019; Zamir et al., 2018; Søgaard & Goldberg, 2016; Hashimoto et al., 2017; Fan et al., 2022; Ye & Xu, 2023; Chen et al., 2023a). Recently, more deep learning models expect integrating different modal and different tasks into one unimodal network (Su et al., 2020; Cho et al., 2021; Hu & Singh, 2021; Lu et al., 2019; Akbari et al., 2021). Their basic motivation is to borrow knowledge or information from the diverse modalities or tasks to help each other. For instance, VATT (Akbari et al., 2021) uses a shared model on video, audio, and text data to perform audio-only, video-only, and image-text retrieval tasks, and HighMMT (Liang et al., 2022) explores modalities beyond the old-school studies of language, vision, and audio to other common modalities such as tabular, time-series, sensors, graphs, and set data, in a multi-task environment. However, there is

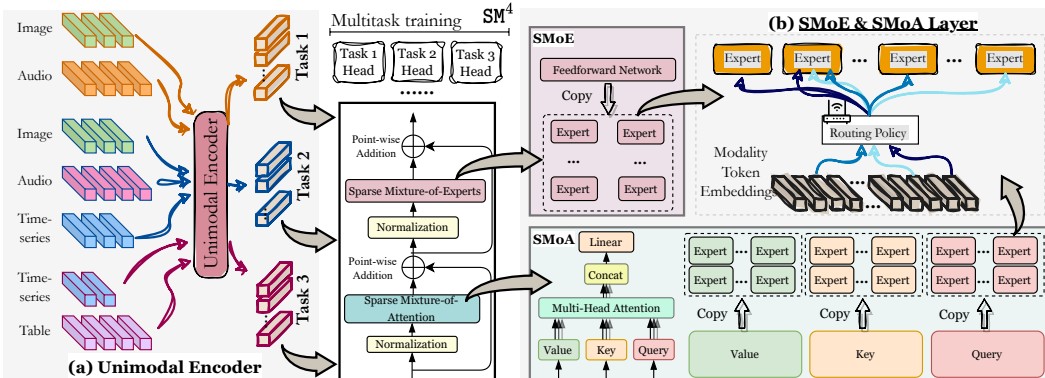

Figure 1: The overall procedure of the proposed framework SM⁴: (a) *Unimodal Encoder.* SM⁴ first stan­dardizes each modality into a sequence, and the unimodal encoder converts each sequence to sequences of the same length. We concatenate these modality tokens on the sequence dimension within each task. Then, the transformer layers of SM⁴ are performed for multi-task learning. (b) *SMoE & SMoA Layer.* Our SM⁴ involves replacing FFN and MSA modules in transformers with SMoE layers and sparse mixture-of-attention (SMoA) layers to split network parameters that mitigate gradient conflict among different modalities and tasks.

no free lunch; unimodal networks introduce more conflicts and complexity during model training. Alamri et al. (2019); Goyal et al. (2017); Poliak et al. (2018); Thomason et al. (2018) show that increasing modalities is not always beneficial. Specifically, the input from different modalities at one optimization object may result in opposite gradient updates (Javaloy et al., 2022; Akbari et al., 2023), which is also observed when inputting the same modality but different learning tasks (Chen et al., 2020). Furthermore, multi-modal networks are often prone to overfitting the easy modalities and impeding performance (Wang et al., 2020a). The various modalities, task resources, and objectives result in unique optimization challenges.

**Sparse Mixture-of-Experts (SMoE).** SMoE as a special instance of conditional computing net­works (Jacobs et al., 1991; Jordan & Jacobs, 1994; Chen et al., 1999; Yuksel et al., 2012), has gained increasing popularity in both vision (Riquelme et al., 2021; Lou et al., 2021; Eigen et al., 2013; Ahmed et al., 2016; Gross et al., 2017; Wang et al., 2020b; Yang et al., 2019; Abbas & Andreopou­los, 2020; Pavlitskaya et al., 2020) and language (Lepikhin et al., 2021; Kim et al., 2021b; Shazeer et al., 2017a; Zhou et al., 2022b; Zhang et al., 2021; Zuo et al., 2022; Jiang et al., 2021) domains. It contains a group of sub-models (*i.e.*, experts) and activates them in an input-dependent fashion. Pioneering investigations leverage its conditional computing nature to assign different model pieces to their most relevant task (Ma et al., 2018b; Aoki et al., 2021; Hazimeh et al., 2021b; Kim et al., 2021a; Fan et al., 2022; Ye & Xu, 2023; Chen et al., 2023a) or modality (Kudugunta et al., 2021; Mustafa et al., 2022) in multi-task or multi-modal learning. To be specific, Ma et al. (2018b); Aoki et al. (2021); Hazimeh et al. (2021b) introduce task-dependent routing policies to select important sub-models given a task and its input sample. Positive results are presented on small-scale uni­modal applications such as classification for medical signal process (Aoki et al., 2021), digital num­ber recognition (MNIST) (Hazimeh et al., 2021b), and recommendation system (Ma et al., 2018b). Mustafa et al. (2022) explores the opportunity of vanilla SMoE in multi-modal contrastive learning. Fan et al. (2022) and Kim et al. (2021a); Rajbhandari et al. (2022); He et al. (2021; 2022) contribute to efficient SMoE frameworks from software-hardware co-design and system angles, respectively.

## 3 METHODOLODY

### 3.1 OVERALL PROCEDURE OF THE SM⁴ FRAMEWORK

The overall procedures of SM⁴ are described in Figure 1. Our proposal processes the multi-task multi-modal learning in a two-step framework. (1) *Unimodal Encoder.* We first process all modali­ties from multi-tasks into sequences; the *Unimodal Encoder* converts each modality into sequences with the same length and concatenates modalities along the sequence dimension for each task. We refer the details of *Unimodal Encoder* to Appendix A.1. (2) *SMoE & SMoA layer.* Then, these se­quential tokens are fed into the transformer layers with SMoE and SMoA, followed by task-specific

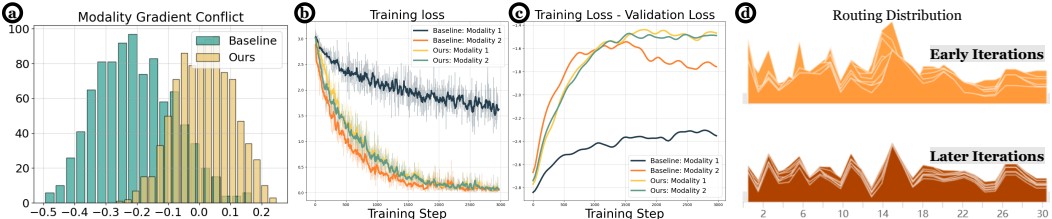

Figure 2: Encompassing comparison between $\text{SM}^4$ with baseline which the dense network with the same Flops. **a**. The distribution of `cosine` distance between training gradients computed from "control" and "propriception" modalities in PUSH dataset. The gradient is collected from the last transformer layer. More positive cosine distances denote less gradient conflict. **b**. The training loss curves, each method collects the loss curve of the "image" and "set" modalities in dataset ENRICO. **c**. The generalization gap of modalities "image" and "set" in dataset ENRICO. A lower generalization gap (the difference between $\text{Loss}_{\text{training}}$, and $\text{Loss}_{\text{valid}}$) indicates better generalization performance (*i.e.*, better modality fitting). **d**. The SMoE routing distribution in dataset ENRICO. Here, we visualize the routing distribution of modality "image" in early and later iterations.

heads. In SMoE and SMoA, routers choose the most relevant experts and aggregate their features for different modalities. The number of selected experts is dynamically decided according to the in-time training dynamics via `AEA`, as detailed below. Each modality's learning pace in one task is adapted via the convergence status of the routing policy from the corresponding modality by `ALP`.

## 3.2 SPARSE MIXTURE OF EXPERTS/ATTENTION IN $\text{SM}^4$

**Sparse Mixture of Experts (SMoE)** SMoE (Shazeer et al., 2017b) has been proposed to enhance model capacity while maintaining low-cost per-inference. In this paper, we use SMoE to disentangle network parameter space for different modalities and tasks. The SMoE layer includes a router network $\mathcal{R}$ and several experts $f_1, f_2, \ldots, f_{\text{E}}$, where E denotes the number of experts. For each input embedding $\mathbf{x}$, $\mathcal{R}$ activates the top-$k$ expert networks with the largest scores $\mathcal{R}(\mathbf{x})_i$, where $i$ is the expert index. The SMoE can be formally denoted as follows:

$$\mathbf{y} = \sum_{i=1}^{k} \mathcal{R}(\mathbf{x})_i \cdot f_i(\mathbf{x}), \mathcal{R}(\mathbf{x}) = \text{TopK}(\text{softmax}(g(x)), k), \tag{1}$$

$$\text{TopK}(\mathbf{v}, k) = \begin{cases} \mathbf{v} & \text{if } \mathbf{v} \text{ is in the top } k \\ 0 & \text{otherwise} \end{cases} \tag{2}$$

where $f_i(\mathbf{x})$ represents the feature produced by expert $f_i$, which is weighted by $\mathcal{R}(\mathbf{x})_i$ to form the final output $\mathbf{y}$. $g$ is the learnable network within a router $\mathcal{R}$, and is commonly is a small FNN with one to few (Shazeer et al., 2017b; Riquelme et al., 2021). `TopK` sets all vector elements to zero except the elements with the largest $k$ values. In $\text{SM}^4$, we duplicate the feedforward network as SMoE expert networks shown in Figure 1 (b).

**Sparse Mixture of Attention (SMoA)** We denote the Mixture-of-Experts in the multi-head self-attention (MSA) module as Sparse Mixture-of-Attention (SMoA). As depicted in Figure 1 (b), we replicate layers to establish expert networks that generate query, key, and value features. Each group of experts in $\text{SM}^4$ is equipped with its own routing policy for outputting queries, keys, or values.

The SMoE and SMoA modules separate network parameter space sufficiently for different modalities and tasks. As supported by our experiments in Section 4, this model architecture mitigates gradient conflicts and enhances performance, which is more suitable for M³TL. More details about SMoE and SMoA, please refer to Appendix A.6

## 3.3 ROUTING POLICY DESIGN IN $\text{SM}^4$

**Routing Policy** Handling multiple modalities without conflicting gradients by disentangling parameters intrinsically relies on a successful routing policy. In $\text{SM}^4$, the routing policy used within expert groups is shared between tasks and modalities. Specifically, for all modalities and tasks, there are four routing policies among SMoE and SMoA. One uses SMoE, and three use SMoA (query,

key, and value routing policies, respectively). Formally, the routing policy for the modality $j$ is:

$$\mathcal{R}_j(\mathrm{x}) = \mathrm{TopK}(\mathrm{softmax}(g(x)), k_j), \qquad (3)$$

where $k_j$ is the modality-specific number of activated experts, and the network $g$ of the router is shared across all modalities and tasks. The routing policy frequently assigns large weights to the same few experts. To combat this imbalance loading phenomenon (Chi et al., 2022), we implement the load and importance balancing loss following Shazeer et al. (2017b). This effective routing policy sends modality embeddings to compatibility experts, which generate high-quality modality features. This helps to solve tasks and separate the network parameter space of different modalities and tasks. As supported by Figure 2 **a**, the disentangled model parameter space results in effective minimized gradient conflict between modalities, enjoying an improved performance (Section 4).

While SMoE and SMoA offer some benefits alone, they are not the silver bullet for multi-modal multi-task learning. Two issues still persist: ❶ *Modality Fitting Issue.* The fixed model capacity in classical SMoE design possibly leads to uneven fitting speeds across modalities. ❷ *Heterogeneous Learning Pace.* The gigantic discrepancy between tasks and modalities can lead to challenges in convergence and optimization pace. Targeting these two obstacles, we propose two solutions that concentrate on SMoE routing and training optimization.

**Adaptive Expert Allocation (`AEA`)**  The optimal fitting pace for each modality may alter significantly due to the difference in modality complexities (Wang et al., 2020a). In $\mathrm{SM}^4$, we use modality-specific $k_j$ to determine the network size of each modality. However, as the number of tasks and modalities increases, computing $k_j$ manually induces high training costs and potential errors (inappropriate $k_j$ can exacerbate overfitting or underfitting for each modality).

Therefore, we adopt an automatic algorithm `AEA` to determine an appropriate $k$ for specific modalities in a data-driven manner. As shown in Figure 2 **c**, we can tune $k_j$ according to the modality-specific validation loss. When the validation loss stops decreasing, we increase the activated model size by increasing $k$. After several training iterations, if the validation loss is still larger than the previous best validation loss, we reduce the selected expert number $k$ for the modality. Ultimately, the modality-specific $k$ is adopted until the end of training. We show the details of `AEA` in Algorithm 2. Figure 2 **c** shows the `AEA` effectively addressing the *Modality Fitting Issue*.

**Adaptive Learning Pace (`ALP`)**  The remaining convergence and optimization pace asynchronization is addressed by our proposed `ALP`. As observed in Figure 2 **d**, the modality-specific routing policy status is unstable in early training iterations and stabilizes in later iterations. Therefore, we monitor the routing distribution entropy as an indicator of routing policy status and, accordingly, decay the learning rate where the modality-specific routing policy entropy is high. As shown in Figure 2 **b**, `ALP` lets us align different learning paces between modalities, which synchronizes the optimization of multiple objectives. Please refer to the details of `ALP` in Algorithm 3.

## 4    EXPERIMENTS

### 4.1    IMPLEMENTATION DETAILS

**Datasets and Tasks.**    To evaluate the proposed method, we conduct experiments on MultiBench, a large-scale multi-modal multi-task benchmark containing more than 10 modalities and 20 prediction tasks across 6 research areas. As shown in Table 1, we follow the HighMMT choose 7 tasks in MultiBench and train 3 multi-modal multi-task models from the combinations of these tasks for the small, medium, and large settings, respectively. For more details, see Appendix C.

**Baselines and Configuration Details.** We consider two state-of-the-art (SOTA) baselines in multi-modal multi-task learning: MultiBench (Liang et al., 2021) and HighMMT (Liang et al., 2022). Particularly, the released code of HighMMT is implemented to achieve the desired performance with provided hyperparameters. MultiBench contains 20 different models for every task; we report the performance range of these models for each adopted task. We display our model architecture overview in Figure 1. We conduct all of our experiments on the NVIDIA A30 Tensor Core GPU. Please refer to Appendix A.5 for more details on network configuration and training setup.

**Evaluation Metrics.**    We use the standard evaluation metrics provided by MultiBench Liang et al. (2021). Specifically, following Vandenhende et al. (2022), we use metric $\Delta$ to evaluate

Table 1: We follow the setting of HighMMT Liang et al. (2022), which uses 3 multi-model multi-task training to evaluate the performance of the $\text{SM}^4$. These setups include tasks with different modality inputs, predicting objectives, research areas, and dataset size.

| Setting | Dataset | Modalities | Prediction Task | Research Area | Size |
|---|---|---|---|---|---|
| Small | PUSH | image,force,proprioception,control | object pose | Robotics | $37,990$ |
| | V&T | image,force,proprioception,depth | contact | Robotics | $147,000$ |
| Medium | ENRICO | image,set | design interface | HCI | $1,460$ |
| | PUSH | image,force,proprioception,control | object pose | Robotics | $37,990$ |
| | AV-MNIST | image,audio | digit | Multimedia | $70,000$ |
| Large | UR-FUNNY | text,video,audio | humor | Affective Computing | $16,514$ |
| | MOSEI | text,video,audio | sentiment | Affective Computing | $22,777$ |
| | MIMIC | time-series,table | ICD-9 codes | Healthcare | $36,212$ |
| | AV-MNIST | image,audio | digit | Multimedia | $70,000$ |

Table 2: Performance comparison, parameter usage, and FLOPS of our model, HighMMT (SOTA multi-modal multi-task learning method on MultiBench benchmark), and all the 20 models implemented in MultiBench (their performance ranges are reported for each dataset) in three settings.

| Setting | Method | Dataset | Performance | $\Delta(\%)$ | # Parameters (M) | FLOPS (G) |
|---|---|---|---|---|---|---|
| Small | MultiBench Models | PUSH↓ | $0.574 \sim 0.290$ | - | $1.09 \sim 135$ | $5.20 \sim 25.11$ |
| | | V&T | $93.30 \sim 93.60$ | | | |
| | HighMMT | PUSH↓ | $0.445$ | $0.00$ | $0.89$ | $5.14$ |
| | | V&T | $96.10$ | | $0.85$ | $32.48$ |
| | $\text{SM}^4$ | PUSH↓ | $0.331$ | $12.93$ | $0.27$ | $2.59$ |
| | | V&T | $96.33$ | | $0.25$ | $17.38$ |
| Medium | MultiBench Models | ENRICO | $44.40 \sim 51.00$ | - | $0.14 \sim 525.70$ | $0.25 \sim 314.13$ |
| | | PUSH↓ | $0.574 \sim 0.290$ | | | |
| | | AV-MNIST | $68.50 \sim 72.80$ | | | |
| | HighMMT | ENRICO | $53.10$ | $0.00$ | $0.58$ | $79.48$ |
| | | PUSH↓ | $0.600$ | | $0.63$ | $21.60$ |
| | | AV-MNIST | $68.48$ | | $0.52$ | $0.95$ |
| | $\text{SM}^4$ | ENRICO | $\mathbf{71.58}$ | $20.19$ | $1.23$ | $1.10$ |
| | | PUSH↓ | $0.475$ | | $1.25$ | $2.33$ |
| | | AV-MNIST | $71.86$ | | $1.23$ | $0.41$ |
| Large | MultiBench Models | UR-FUNNY | $60.20 \sim 66.70$ | - | $0.19 \sim 31.50$ | $0.15 \sim 21.60$ |
| | | MOSEI | $76.40 \sim 82.10$ | | | |
| | | MIMIC | $67.60 \sim 68.90$ | | | |
| | | AV-MNIST | $65.10 \sim 72.80$ | | | |
| | HighMMT | UR-FUNNY | $62.00$ | $0.00$ | $0.52$ | $1.51$ |
| | | MOSEI | $78.40$ | | $0.52$ | $1.65$ |
| | | MIMIC | $65.60$ | | $0.52$ | $0.67$ |
| | | AV-MNIST | $70.60$ | | $0.52$ | $0.95$ |
| | $\text{SM}^4$ | UR-FUNNY | $64.24$ | $2.28$ | $0.76$ | $0.38$ |
| | | MOSEI | $79.47$ | | $0.76$ | $0.53$ |
| | | MIMIC | $67.91$ | | $0.76$ | $0.15$ |
| | | AV-MNIST | $71.05$ | | $0.76$ | $0.43$ |

the performance gap between our model and baseline averaged over all the tasks in each set: $\Delta = \frac{1}{T}\sum_i^T (-1)^{l_i}(M_{m,i} - M_{b,i})/M_{b,i}$, where $M_{m,i}$ and $M_{b,i}$ denote the performances of our $\text{SM}^4$ and baseline model, respectively; $T$ is the number of considered tasks; and $l_i = 1$ if a higher metric value means better performance otherwise $l_i = -1$. The results of HighMMT and $\text{SM}^4$ are reported by the mean of three independent runs. For the min and max performances of MultiBench, we reuse the numbers directly from its publication to have a comprehensive comparison.

## 4.2 PERFORMANCE COMPARISON OF $\text{SM}^4$ WITH EXISTING MULTIMODEL MODELS

We compare our model's performance with SOTA HighMMT Liang et al. (2022) as well as 20 multi-modal models implemented in benchmark MultiBench Liang et al. (2021). The comparison results are collected in Table 2, from which we make the following observations. ① Our $\text{SM}^4$ demonstrates

Table 3: Comparison of routing design. $\text{SM}^4$ makes use of the single router; Multi-router, R-Multi-router, P-Modality-router, and P-Task-router mean the adoptions of task-specific and/or modality-specific routing networks in SMoE and SMoA, respectively. Further investigations of the combinations between the multi-routing networks and the single-routing networks are in Appendix B.

| Model | ENRICO ↑ | PUSH ↓ | AV-MNIST ↑ | $\Delta(\%)$ ↑ |
|---|---|---|---|---|
| HighMMT multitask | 53.10 | 0.600 | 68.48 | 0.00 |
| $\text{SM}^4$ (ours) | **71.58** | **0.475** | **71.86** | **20.19** |
| Multi-router $\text{SM}^4$ | 71.00 | 0.684 | 71.03 | 7.81 |
| R-Multi-router $\text{SM}^4$ | 64.38 | 0.995 | 71.33 | -13.48 |
| P-Modality-router $\text{SM}^4$ | 68.72 | 0.786 | 70.70 | 0.54 |
| P-Task-router $\text{SM}^4$ | 68.38 | 0.833 | 70.69 | -2.25 |

great advantages with a clear performance margin compared to all baselines. Specifically, compared to the multi-modal multi-task model HighMMT, $\text{SM}^4$ achieves improvements up to 12.93%, 20.19%, and 2.28% for small, medium, and large settings, respectively. These empirical results validate the effectiveness of our model to address the cross-task conflict and assign expert sub-networks to conduct each prediction task. ② $\text{SM}^4$ adaptively allocates adequate amounts of model parameters and fewer FLOPS to resolve the different tasks. For example, our method uses fewer parameters compared to HighMMT in the easy, small setting, e.g., $1.38\% \sim 53.51\%$ parameter saving, while we use larger parameter budgets in the challenging medium and large settings. The required FLOPS of $\text{SM}^4$ is always smaller than that of HighMMT. In other words, we have more efficient inference per task. ③ $\text{SM}^4$ delivers significant improvements and creates SOTA performances for some tasks. Notably, at the prediction task on ENRICO, $\text{SM}^4$ obtains 20.58% improvement compared with the best-performing model on MultiBench.

## 4.3 DETAILED INVESTIGATIONS OF $\text{SM}^4$

**Ablation Study: Single-router v.s. Multi-router.** Unlike the routing policy design in $\text{SM}^4$, we notice that earlier works have investigated task-specific or modality-specific routing networks in learning the routing policy individually for different modalities or tasks in MTL Ma et al. (2018b); Aoki et al. (2021); Hazimeh et al. (2021b); Kim et al. (2021a). Therefore, we ask ***What kind of routing policy is suitable for $M^3TL$?*** Under the medium setting with framework $\text{SM}^4$, we experiment with 4 multi-router designs to identify the optimal routing policy. We use modality-specific routers in SMoA and task-specific routers in SMoE, which are named as Multi-router $\text{SM}^4$. Alternatively, in R-Multi-router $\text{SM}^4$, we utilize modality-specific routers in SMoE and task-specific routers in SMoA. In SMoA and SMoE, we employed task-specific routers as P-Task-router $\text{SM}^4$, and modality-specific routers as P-Modality-router $\text{SM}^4$, respectively. From our results in Table 3, ① we observe that the adopted single router consistently outperforms the other routing policies. ② Specifically, all the task-specific routers perform unpromisingly in PUSH, which contains four different data modalities. We extend this and draw similar conclusions in Appendix B.

Table 4: Ablation of SMoE and SMoA. Notably, the "Dense Model" has the same computation cost with $\text{SM}^4$. The results of the dense model with the same network capacity are in Appendix B.

| Model | ENRICO ↑ | PUSH ↓ | AV-MNIST ↑ | $\Delta(\%)$ ↑ |
|---|---|---|---|---|
| HighMMT multitask | 53.10 | 0.600 | 68.48 | 0.00 |
| $\text{SM}^4$ (ours) | **71.58** | **0.475** | **71.86** | **20.19** |
| - w/o SMoA | 69.06 | 1.227 | 70.26 | −23.92 |
| - w/o SMoE | 68.84 | 0.818 | 70.94 | −1.02 |
| Dense Model | 65.98 | 1.342 | 70.49 | −32.14 |

**Ablation Study: MoE.** To investigate the contribution of MoE, the ablation studies are conducted with $\text{SM}^4$ on the medium setting. In particular, we consider three ablated models: *$SM^4$ w/o SMoA*: removing SMoA from the MSA module. *$SM^4$ w/o SMoE*: removing SMoE from FFN layer. *Dense model*: using the same computation cost with $\text{SM}^4$ but without any MoE components. From Table 3, we make the following observations: ① Compared with $\text{SM}^4$, the ablation of any MoE component

Table 5: **Ablation studies**. Hyperparameter effects of the total number of experts (i.e., $N$) on $\text{SM}^4$.

| Model | ENRICO ↑ | PUSH ↓ | AV-MNIST ↑ | $\Delta(\%)$ ↑ |
|---|---|---|---|---|
| HighMMT multitask | 53.10 | 0.600 | 68.48 | 0.00 |
| $N = 32$ ($\text{SM}^4$) | **71.58** | **0.475** | **71.86** | 20.19 |
| $N = 4$ | 67.92 | 1.250 | 71.33 | $-25.41$ |
| $N = 8$ | 67.69 | 0.975 | 70.93 | $-10.51$ |
| $N = 16$ | 69.75 | 0.771 | 70.45 | 1.89 |

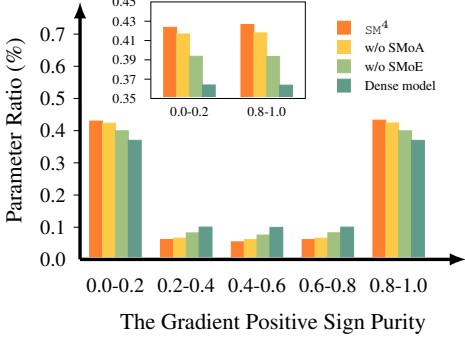
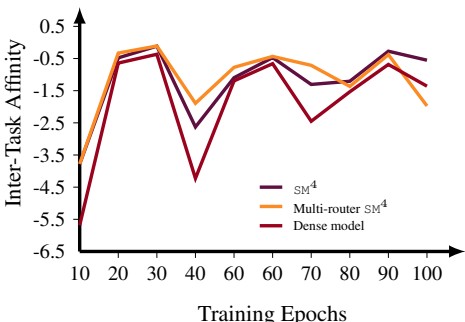

Figure 3: The distribution of Gradient Positive Sign Purity (left), and the inter-task affinity of the 'ENRICO' to the 'PUSH' task (right).

significantly discounts model performance. Specifically, discarding SMoA has a more acute drop compared to discarding SMoE. This verifies our motivation for applying SMoA, which improves the model's routing capability. ② Compared with the dense model, $\text{SM}^4$ achieves a noticeable performance gain (*e.g.*, $\geq 1.37\%$), suggesting the benefit from the distanglement of parameter spaces.

**Ablation Study: Expert counts.** For the SMoE and SMoA layers, the total number of experts $N$ is one of the most significant hyper-parameters. We show the detailed performance in Table 5 and observe that increasing $N$ improves model performance but costs more required memory. Choosing $N$ appropriately is crucial for $\text{SM}^4$.

**In-Depth Discussion: Do our proposals address the gradient conflict between modalities and tasks?** Yes, $\text{SM}^4$ is specialized to disentangle the task conflict by harmonizing the updating gradient of different tasks. We examine the following two metrics.

▷ *Gradient positive sign purity (GPSP).* This metric quantifies the direction consistency of backward gradients of different tasks Chen et al. (2020). Mathematically, we denote GPSP as $\mathcal{P}$ and record the gradient of task $i$ as $\nabla \mathtt{W}_i$. Metric GPSP is defined as $\mathcal{P} = \sum_i \nabla \mathtt{W}_i / \sum_i |\nabla \mathtt{W}_i|$, which is further bounded into range $[0, 1]$. Specifically, $\mathcal{P}$ with a value closing to 0 or 1 indicates that the gradients from different tasks are not acutely contradictory to each other. We compare GPSP distributions of $\text{SM}^4$, $\text{SM}^4$ without MoE on self-attention, $\text{SM}^4$ without MoE on FFN, and the dense model. In Figure 3, we discretize the values of $\mathcal{P}$ into 5 intervals and then count the number of parameters that fall within each interval. Compared with other models, the GPSP values of $\text{SM}^4$ are accumulated more at the intervals of $[0.6, 0.8]$ and $[0.8, 1.0]$. This validates the effectiveness of splitting the parameter space, where only a small fraction of conflicting parameters are running for specific tasks.

▷ *Inter-task affinity.* We denote inter-task affinity with $Z_{i \to j}$, which is the influence of parameter update from task $i$ to task $j$ Fifty et al. (2021). The higher value of $Z_{i \to j}$ means the parameter update is positive for task $j$; otherwise, task $j$ suffers from an antagonistic updating. On the medium setting, we compare the inter-task affinity of task ENRICO to task PUSH for three backbones: $\text{SM}^4$, multi-router $\text{SM}^4$, and dense model. As shown in the right part of Figure 3, we observe the inter-task affinities of $\text{SM}^4$ and multi-router $\text{SM}^4$ tend to be higher than that of the dense model. This finding shows that MoE can restrain the gradient conflict of MTL. For more discussions on GPSP and inter-task affinity, please refer to Appendix C.4 and Appendix C.5.

**In-depth Discussion: Whether our proposals address the fitting issue between modalities?** Yes, we examine this question by visualizing the training loss dynamic (second subfigure) and generalization gap dynamic (third subfigure) in Figure 2. Note that the generalization gap is defined by the

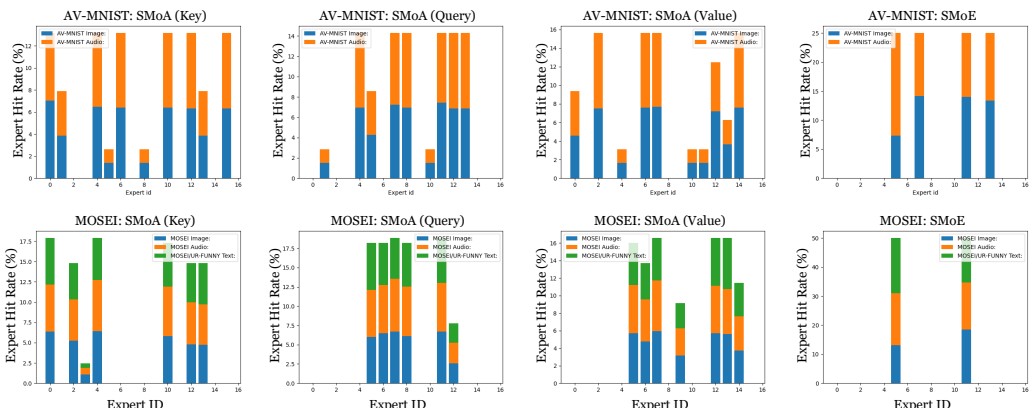

Figure 4: Analysis on the expert selection visualization produced by the $SM^4$ in the large setting. The first row shows the expert selection in the "AV-MNIST" dataset, and the second row shows the expert selection in the "MOSEI" dataset. For more results about expert selection, please refer to Appendix C.7.

difference between training loss and validation loss, where a higher value means a good generalization performance on the validation set. ① *It is observed baseline HighMMT underfits in specific modality, which has the highest training loss accompanied with a lower generalization gap.* In contrast, another modality is gradually overfitted along with the training process in HighMMT. ② $SM^4$ *delivers comparably superior results in all the modalities, which addresses the key challenge of under/over-fitting in multi-modal learning.* $SM^4$ consistently outperforms highMMT by obtaining superior generalization gaps in all modalities.

**In-depth Discussion: Is the expert selection specialized to the different modalities and tasks?** We show the routing distributions for different modalities and tasks of the medium setting in Figure 4, from which we make the following observations. ① *There is an overall balanced loading across the different modalities in SMoA, but it shows an imbalance in some of the experts in SMoE.* For example, expert 5 prefers modality audio in the AV-MNIST task and prefers modalities of audio as well as text in the MOSEI task. ② *The expert selection is specialized to the different tasks.* Considering the SMoA (Query) layer, we observe the AV-MNIST task leverages the unique experts 4 and 13 while the MOSEI task activates expert 6. These empirical studies show $SM^4$ can optimize how many (*i.e.*, adaptive network capacity) and which (*i.e.*, dynamic routing) experts to activate for each task and modality.

## 5 CONCLUSION AND LIMITATION

This paper introduces $SM^4$, using Sparse Mixture-of-Experts to address the forgetting, fitting, and learning issues in multi-modal multi-task learning. By tailoring the Mixture-of-Experts into both the self-attention and the feed-forward networks of a transformer backbone, we achieve the following. First, the Sparse Mixture-of-Attention (SMoA) and the Sparse Mixture-of-Experts (SMoE) sufficiently disentangle the network parameter space to mitigate the gradient conflict between different modalities and tasks. Second, we design an adaptive expert allocation mechanism to determine the number of selected experts in use for different modalities, resulting in unified fitting speeds between modalities. Third, we adapt the learning pace by considering the convergence status of modality-specific routing policies to synchronize the learning paces of different modalities and tasks. Comprehensive experiments show that the proposed $SM^4$ surpasses the SOTA with a fraction of the computation cost (+12.93%/+20.19%/+2.28% $M^3TL$ performance); our computation cost is only 1.38% ∼ 53.51% of the SOTA model. Our experiments on MoE also provide rational perspectives for designing multi-modal multi-task learning neural network architectures. The limitation of our work is that the proposed $SM^4$ is only evaluated on academic datasets. Moving forward, we will evaluate $SM^4$ on more practical tasks like in-door robots and autonomous vehicles. Also, we expect to expand our model size for larger-scale tasks and more kinds of modalities in future work.

## REPRODUCIBILITY STATEMENT

The authors have taken great care to ensure the reproducibility of algorithms and results presented in the paper. Section 4 and Appendix C provide detailed information about the experimental settings. This paper analyzes 7 datasets within one benchmark, with comprehensive information about each dataset available in Table 1 and Appendix C.1. Additionally, the evaluation metrics have been explained in Section 4, offering a clear framework for assessing the results of the proposed method. We highlight that our performance of HighMMT is produced with the official implementation from HighMMT's repository (HightMMT). We strictly follow the default configurations reported in their paper, as shown in Tables 8, 9, and 10. For example, we use learning rates of 0.0005, 0.001, and 0.0008 for the small, medium, and large settings, respectively. Additionally, supplementary material includes codes and reproduction scripts of SM[4] and HighMMT.

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

# A MODEL DETAILS

## A.1 PROCESS DATA INTO SEQUENCE

Following the process of Jaegle et al. (2022), we first standardize each input into a sequence. For each modality Jaegle et al. (2022), we define some hyperparameters (such as max_freq, num_freq_bands, and freq_base) for the Fourier positional encoding. Fourier transformations get this positional information. For modalities such as text and time-series, they are already sequential data. We apply 1D positional encoding for these modalities $x \in \mathbb{R}^{b_m \times t_m \times d_m}$, where $b_m, t_m, d_m$ are the batch size, sequence length, and input dimension of current modality, respectively. For image and similar modalities, we follow the processing procedure of Dosovitskiy et al. (2021), which breaks each input into $h_m \times w_m$ patches and flattens it as a sequence of $p^2$ regions. We use 2D positional encoding for image and similar modalities input $x \in \mathbb{R}^{b_m \times h_m \times w_m \times d_m}$, where $h_m \times w_m$ is the number of patches. For image modality, the $d_m$ is the number of pixels within a patch. For video and similar modalities, we treat each frame data as the image modality, therefore we apply 3D positional encoding for input $x \in \mathbb{R}^{b_m \times l_m \times h_m \times w_m \times d_m}$, where $l_m$ is the number of the frame. In the other modalities, such as table and graph, we treat each element in the table/graph as an element in the sequence and use a 1D positional encoding.

After transposing inputs into sequence data, we show the subsequent processing procedure in Algorithm 1. The 'max_modality_dim' equals to $\max_{m \in M}(d_m + d_{pm})$, where $d_{pm}$ is the dimension of Fourier positional encoding for the corresponding modality. The one-hot encoding is defined as $e_m \in \mathbb{R}^{|M|}$, where $|M|$ is the number of all modalities involved.

---

**Algorithm 1** Data Preprocess in Python style

---

```python
# x: the input tokens of specific modality
def data_preprocess(x,modality, max_modality_dim):
    # get positional encoding information
    # pos_dim: indicates 1D/2D/3D positional encoding
    enc_pos = fourier_encode(modality.pos_dim,
                    modality.max_freq,
                    modality.num_freq_bands,
                    modality.freq_base)
    # add padding for modalities with smaller input dimension
    # max_modality_dim: the maximum input dimension overall modalities
    # input_dim: the input dimension of the current modality
    padding=zeros(max_modality_dim-modality.input_dim)
    # modality one-hot encoding
    # modality_index: the index of current modality
    modality_encodings = one_hot(modality.modality_index)
    # construct final input
    modality_input = concatenate(x, padding, enc_pos, modality_encodings)
    return modality_input
```

---

## A.2 THE UNIMODAL ENCODER

The result of Algorithm 1 is then fed into the unimodal encoder layer. We display the details of the unimodal encoder layer in Figure 5. The sequence length $T$ of different modalities are different, as $T$ can be $t_m$, $h_m \times w_m$, or $l_m \times h_m \times w_m$. However, the cross-attention between the input sequence and latent input will convert the sequence length from different modalities into the same value. For example, the input modality sequence is $x \in \mathbb{R}^{T_m \times D}$ and the latent input is $z \in \mathbb{R}^{N \times C}$. After these three linear layer, we got $\mathbf{K}, \mathbf{V} \in \mathbb{R}^{T_m \times X}$ and $\mathbf{Q} \in \mathbb{R}^{N \times X}$. Following the scaled-dot product attention:

$$Attention(\mathbf{Q}, \mathbf{K}, \mathbf{V}) = softmax(\frac{\mathbf{Q}\mathbf{K}^T}{\sqrt{C}})\mathbf{V}, \tag{4}$$

from which we can know the dimension after the attention is $Attention(\mathbf{Q}, \mathbf{K}, \mathbf{V}) \in \mathbb{R}^{N \times X}$. Therefore, the sequence length of the output depends on the sequence length of the latent input, and the feature dimension depends on the unimodal encoder's hidden size, which is independent of the shape

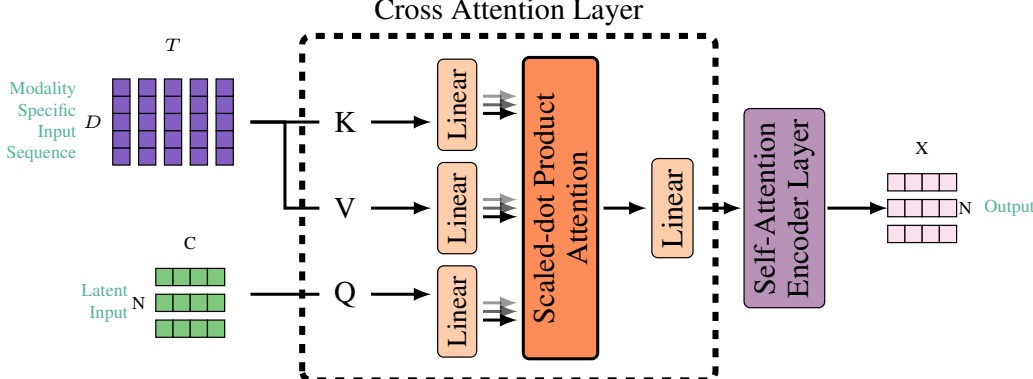

Figure 5: The details of the unimodal encoder layer. The $D$ and $T$ are the sequence length and feature dimension of the modality-specific input sequence. The $N$ and $C$ are the sequence length and the number of dimensions of latent input. The latent input is the learnable parameters shared across different modalities and tasks.

of the input modality sequence. The hidden dimension of the self-attention encoder layer equals the previous layer's cross-attention layer.

### A.3 DETAILS OF ADAPTIVE EXPERT ALLOCATION

The python style pseudo code for the Adaptive Expert Allocation (AEA) algorithm 2. The AEA is executed during the multi-modal multi-task learning.

---

**Algorithm 2** Adaptive Expert Allocation (AEA) in Python style

```python
def adaptive_expert_allocation(modality_set, model, modality_topk):
    for modality in modality_set:
        improved = True
        loss_val = inf
        while True:
            # if the valid loss does not decrease iterations
            if loss_decrease(loss_val):
                if not improved:
                    break
                else:
                    n_experts = n_experts + 1
                    improved = False
            # training model 1 epoch
            loss_val_i = train(model)
            if loss_val_i < loss_val:
                loss_val = loss_val_i
                improved = True
        n_experts = n_experts - 1
        # get the top-k expert number
        modality_topk[modality]=n_experts
        continue_train(model) # continue training the target number of
            epochs
    return modality_topk
```

---

### A.4 DETAILS OF ADAPTIVE LEARNING PACE

The python style pseudo code for the Adaptive Learning Pace (ALP) algorithm 3. The ALP adjusts the learning pace of each modality by out modality-specific learning rate weights.

**Algorithm 3** Adaptive Expert Allocation (`AEA`) in Python style

```python
def adaptive_learning_pace(modality_set, model):
    entropys = []
    for modality in modality_set:
        entropys.append(routing_entropy(model, modality))
    return 1 - entropys.softmax()
```

## A.5 THE MODEL AND TRAINING SETUPS

We list hyperparameters for the training and the model in Table 14, Table 15, and Table 16 for small, medium and large settings, respectively.

## A.6 EXPERT GROUP

Our framework comprises four distinct expert groups: one group, situated within SMoE, consists of experts duplicating feedforward networks. Meanwhile, SMoA includes three expert groups, each dedicated to duplicating the query, key, and value networks, respectively. Experts are grouped by the nature of where they are duplicated from.

## B SINGLE-ROUTER V.S. MULTI-ROUTER IN SM$^4$

SM$^4$ use SMoE and SMoA to disentangle the network parameter space. Moreover, several works Ma et al. (2018b); Aoki et al. (2021); Hazimeh et al. (2021b); Kim et al. (2021a); Kudugunta et al. (2021); Mustafa et al. (2022); Kim et al. (2021a) investigate single-router or multi-router for multi-task learning or multi-modal learning. Therefore, we also investigate the multi-router SM$^4$ for M$^3$TL. With those in mind, we ask a much more significant question:

*What kind of router design is appropriate for SM$^4$ to M$^3$TL?*

For our proposed SM$^4$, we can use Single-router and Multi-router in both the self-attention and FFN layers, respectively. Meanwhile, the Multi-router can also be divided into the modality-specific Multi-router and the task-specific Multi-router. Therefore, we explore all possible combinations of the above settings SMoE and SMoA. Note that, without specifics, the router in SMoE and SMoA is a single router by default. Herein, the "single router" denotes one router in SMoE and three routers in SMoA ("single" refers to not using task/modality-specific in SMoE or SMoA). We list all explored network architectures in Table 6.

Table 6: All possible router design combinations for SM$^4$.

| | | SMoE | | | |
| --- | --- | --- | --- | --- | --- |
| | | Modality-Specific Router | Task-Specific | Single-Router | $w/o$ SMoE |
| | Modality-Specific Router | P-Modality-router SM$^4$ | Multi-router SM$^4$ | Modality-SMoA-Single-SMoE SM$^4$ | Multi-router SM$^4$ $w/o$ SMoE |
| SMoA | Task-Specific Router | R-Multi-router SM$^4$ | P-Task-router SM$^4$ | Task-SMoA-Single-SMoE SM$^4$ | R-Multi-router SM$^4$ $w/o$ SMoE |
| | Single-Router | Single-SMoA-Modality-SMoE SM$^4$ | Single-SMoA-Task-SMoE SM$^4$ | SM$^4$ | SM$^4$ $w/o$ SMoE |
| | $w/o$ MoA | R-Multi-Router SM$^4$ $w/o$ SMoA | Multi-Router SM$^4$ $w/o$ SMoA | SM$^4$ $w/o$ SMoA | Dense Model |

We run the above network architectures in the medium setting and report the results in Table 7. All results reported in Table 7 use the same hyperparameters in Table 15, except for the routing network setting. In particular, the 'Dense Model' is an equal computation dense model where we propose two kinds of equal computation dense model: 'Dense Model 1' uses the transformer encoder layer with double depth, and 'Dense Model 2' is 4x wider than the hidden dimension of the transformer encoder layer. To further illustrate our performance gains mainly come from our SM$^4$ design, we construct the same capacity model where we $\times 4$ the number of attention heads, $\times 8$ the dimension of each attention head, and $\times 32$ the hidden dimension of the FFN layer. .

We find out that the single-router is the best architecture for M$^3$TL. The second-best architecture uses the task-specific router in the SMoE and the dense layer in the FFN layer. Meanwhile, using

Table 7: The results of different SMoE & SMoA router settings in the medium setting.

| Model | ENRICO ↑ | PUSH ↓ | AV-MNIST ↑ | Δ(%) ↑ |
|---|---|---|---|---|
| HighMMT multitask | 53.10 | 0.600 | 68.48 | 0.00 |
| $\text{SM}^4$ | **71.58** | **0.475** | **71.86** | **20.19** |
| Multi-router $\text{SM}^4$ | 71.00 | 0.684 | 71.03 | 7.81 |
| R-Multi-router $\text{SM}^4$ | 64.38 | 0.995 | 71.33 | −13.48 |
| Dense Model 1 | 65.98 | 1.342 | 70.49 | −32.14 |
| Dense Model 2 | 62.56 | 1.400 | 71.40 | −37.11 |
| $\text{SM}^4$ $w/o$ SMoE | 68.84 | 0.818 | 70.94 | −1.02 |
| $\text{SM}^4$ $w/o$ SMoA | 69.06 | 1.227 | 70.26 | −23.92 |
| Multi-router $\text{SM}^4$ $w/o$ SMoE | 67.58 | 1.166 | 71.11 | −21.06 |
| Multi-router $\text{SM}^4$ $w/o$ SMoA | 65.41 | 1.402 | 70.08 | −36.03 |
| R-Multi-router $\text{SM}^4$ $w/o$ SMoE | 67.35 | 0.633 | 71.37 | 8.54 |
| R-Multi-router $\text{SM}^4$ $w/o$ SMoA | 66.43 | 0.969 | 71.04 | −10.89 |
| Task-SMoA-Single-SMoE $\text{SM}^4$ | 63.81 | 0.952 | 71.02 | −11.62 |
| Modality-SMoA-Single-SMoE $\text{SM}^4$ | 69.52 | 0.777 | 71.47 | 1.94 |
| Single-SMoA-Task-SMoE $\text{SM}^4$ | 67.24 | 0.764 | 71.03 | 1.00 |
| Single-SMoA-Modality-SMoE $\text{SM}^4$ | 65.75 | 1.088 | 71.31 | −17.77 |
| P-Modality-router $\text{SM}^4$ | 68.38 | 0.786 | 70.70 | 0.54 |
| P-Task-router $\text{SM}^4$ | 68.38 | 0.833 | 70.69 | −2.25 |
| Equal Capacity Model | 64.61 | 0.878 | 69.80 | −7.59 |

Table 8: Task performances of different models. $\text{SM}^4$ 2/3/4 layers: 2/3/4 transformer encoder layers and replacing with $\text{SM}^4$ layer every other layer. P-$\text{SM}^4$ 2/3/4 layers: 2/3/4 consecutive $\text{SM}^4$ layers. $\text{SM}^4$ early/middle/late-2: 4 transformer encoder layers and replacing the early/middle/late-2 encoder layers with two $\text{SM}^4$ layers.

| Model | ENRICO ↑ | PUSH ↓ | AV-MNIST ↑ | Δ(%) ↑ |
|---|---|---|---|---|
| HighMMT multitask | 53.10 | 0.600 | 68.48 | 0.00 |
| $\text{SM}^4$ | **71.58** | **0.475** | **71.86** | **20.19** |
| $\text{SM}^4$ 2 layers | 70.55 | 0.992 | 70.34 | −9.92 |
| $\text{SM}^4$ 3 layers | 69.18 | 0.551 | 70.32 | 13.71 |
| $\text{SM}^4$ 4 layers | 71.46 | 1.223 | 70.18 | −22.24 |
| P-$\text{SM}^4$ 2 layers | 69.63 | 0.766 | 71.57 | 2.64 |
| P-$\text{SM}^4$ 3 layers | 70.78 | 0.616 | 71.12 | 11.49 |
| P-$\text{SM}^4$ 4 layers | 67.47 | 0.976 | 71.68 | −10.30 |
| $\text{SM}^4$ early two layer | 68.15 | 0.793 | 71.19 | −0.03 |
| $\text{SM}^4$ middle two layer | 73.17 | 0.884 | 69.89 | −2.49 |
| $\text{SM}^4$ late two layer | 72.15 | 1.374 | 69.97 | −30.33 |

the modality-specific router in the SMoA and the task-specific router in SMoE also seems like a reasonable choice.

For better understanding, we display the architecture of the **Multi-Router $\text{SM}^4$** and the **R-Multi-Router $\text{SM}^4$** in Figure 6 and Figure 7, respectively.

### B.1 USING CONSECUTIVE $\text{SM}^4$

This section is used to illustrate how to use the consecutive $\text{SM}^4$ layer (*i.e.*, transformer layer with SMoE and SMoA design) as transformer backbone and provide more observation about how to use $\text{SM}^4$ while the network is getting deeper.

Our experimental results in Table 8 show:

- The performance may not be improved as the number of $\text{SM}^4$ layers increases.
- The location of $\text{SM}^4$ matters. Using $\text{SM}^4$ in shallow layers helps the most.

## C EXPERIMENTS DETAILS

We show the number of parameters and the computation cost of the current SOTA and $\text{SM}^4$ in Figure 9. The "small", "medium", and "large" setting denote the number of tasks.

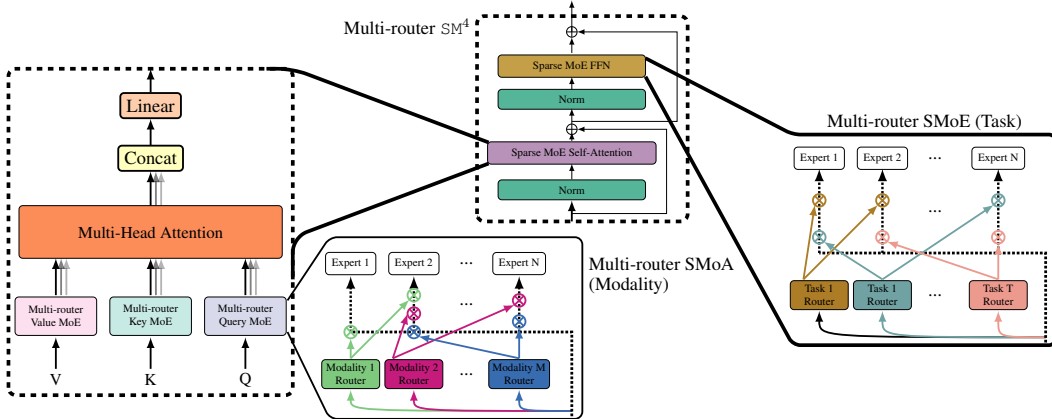

Figure 6: In the **Multi-router SM**$^4$ encoder layer, We use the modality-specific router in the SMoA and the task-specific router in the SMoE.

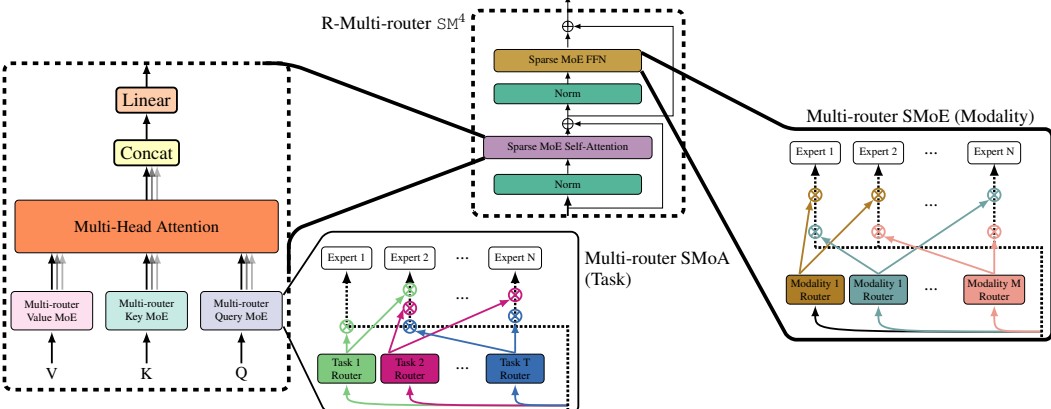

Figure 7: In the **Reverse Multi-router SM**$^4$ (**R-Multi-router SM**$^4$) encoder layer, We use the task-specific router in the SMoA and the modality-specific router in the SMoE.

Table 9: Detailed results of parameter and computation cost.

| Small setting | PUSH | | V&T | |
|---|---|---|---|---|
| | Params (M) | Flops (G) | Params (M) | Flops (G) |
| HighMMT multitask | 0.89 | 5.14 | 0.85 | 32.48 |
| SM$^4$ | 0.27 | 2.59 | 0.25 | 17.38 |

| Medium setting | ENRICO | | PUSH | | AV-MNIST | |
|---|---|---|---|---|---|---|
| | Params (M) | Flops (G) | Params (M) | Flops (G) | Params (M) | Flops (G) |
| HighMMT multitask | 0.58 | 79.48 | 0.63 | 21.60 | 0.52 | 0.95 |
| SM$^4$ | 1.23 | 1.10 | 1.25 | 2.33 | 1.23 | 0.41 |

| Large setting | UR-FUNNY | | MOSEI | | MIMIC | | AV-MNIST | |
|---|---|---|---|---|---|---|---|---|
| | Params (M) | Flops (G) | Params (M) | Flops (G) | Params (M) | Flops (G) | Params (M) | Flops (G) |
| HighMMT multitask | 0.52 | 1.51 | 0.52 | 1.65 | 0.52 | 0.67 | 0.52 | 0.95 |
| SM$^4$ | 0.76 | 0.38 | 0.76 | 0.53 | 0.76 | 0.15 | 0.76 | 0.43 |

In Table 10, we display more comprehensive performance about MultiBench to help readers locate the position of SM$^4$ in the MultiBench benchmark. For each dataset, we choose multi-modal models with the best/worst performance and multi-modal models with the largest/smallest parameter numbers, respectively.

Table 10: Detailed performance comparison, parameter usage, and FLOPS of our model, HighMMT, SOTA multi-modal multi-task learning method on MultiBench benchmark in three settings. For the "FLOPS(G)", "-" indicates the MultiBench does not provide official implementation. Notably, the empty FLOPS of the "MultiBench Model (MFAS)" is due to the FLOPS of "MFAS" being dynamic during training.

| Setting | Method | Dataset | Performance | $\Delta(\%)$ | # Parameters (M) | FLOPS (G) |
|---|---|---|---|---|---|---|
| Single Task | MultiBench Models (TF-LSTM) | PUSH↓ | 0.574 | – | 23.5 | 25.11 |
| | MultiBench Models (LF-LSTM) | PUSH↓ | 0.290 | – | 1.90 | 14.07 |
| | MultiBench Models (MULT) | PUSH↓ | 0.402 | – | 14.6 | 19.20 |
| | MultiBench Models (LRTF) | V&T | 93.3 | – | 1.09 | 5.20 |
| | MultiBench Models (LF) | V&T | 93.6 | – | 1.20 | 5.20 |
| | MultiBench Models (RefNet) | V&T | 93.5 | – | 135 | – |
| | MultiBench Models (TF) | ENRICO | 46.6 | – | 19.3 | 314.13 |
| | MultiBench Models (GradBlend) | ENRICO | 51.0 | – | 19.3 | 314.13 |
| | MultiBench Models (RefNet) | ENRICO | 44.4 | – | 25.7 | 2.67 |
| | MultiBench Models (GradBlend) | AV-MNIST | 68.5 | – | 0.29 | 0.50 |
| | MultiBench Models (MFAS) | AV-MNIST | 72.8 | – | 0.14 | – |
| | MultiBench Models (RefNet) | AV-MNIST | 70.9 | – | 14.1 | 0.25 |
| | MultiBench Models (EF-GRU) | UR-FUNNY | 60.2 | – | 3.58 | 3.13 |
| | MultiBench Models (MULT) | UR-FUNNY | 66.7 | – | 2.38 | 3.37 |
| | MultiBench Models (MCTN) | UR-FUNNY | 63.2 | – | 0.19 | 0.17 |
| | MultiBench Models (TF) | UR-FUNNY | 61.2 | – | 12.2 | 2.67 |
| | MultiBench Models (MCTN) | MOSEI | 76.4 | – | 0.19 | 0.15 |
| | MultiBench Models (MULT) | MOSEI | 82.1 | – | 4.75 | 3.35 |
| | MultiBench Models (LF-Transformer) | MOSEI | 80.6 | – | 31.5 | 21.60 |
| | MultiBench Models (MI-Matrix) | MIMIC | 67.9 | – | 0.801 | 0.005 |
| | MultiBench Models (LF) | MIMIC | 68.9 | – | 0.034 | 0.005 |
| | MultiBench Models (LRTF) | MIMIC | 68.5 | – | 0.008 | 0.005 |
| Small | HighMMT | PUSH↓ | 0.445 | 0.00 | 0.89 | 5.14 |
| | | V&T | 96.10 | | 0.85 | 32.48 |
| | SM$^4$ | PUSH↓ | 0.331 | 12.93 | 0.27 | 2.59 |
| | | V&T | 96.33 | | 0.25 | 17.38 |
| Medium | HighMMT | ENRICO | 53.10 | 0.00 | 0.58 | 79.48 |
| | | PUSH↓ | 0.600 | | 0.63 | 21.60 |
| | | AV-MNIST | 68.48 | | 0.52 | 0.95 |
| | SM$^4$ | ENRICO | **71.58** | 20.19 | 1.23 | 1.10 |
| | | PUSH↓ | 0.475 | | 1.25 | 2.33 |
| | | AV-MNIST | 71.86 | | 1.23 | 0.41 |
| Large | HighMMT | UR-FUNNY | 62.00 | 0.00 | 0.52 | 1.51 |
| | | MOSEI | 78.40 | | 0.52 | 1.65 |
| | | MIMIC | 65.60 | | 0.52 | 0.67 |
| | | AV-MNIST | 70.60 | | 0.52 | 0.95 |
| | SM$^4$ | UR-FUNNY | 64.24 | 2.28 | 0.76 | 0.38 |
| | | MOSEI | 79.47 | | 0.76 | 0.53 |
| | | MIMIC | 67.91 | | 0.76 | 0.15 |
| | | AV-MNIST | 71.05 | | 0.76 | 0.43 |

## C.1 DATASET

**PUSH** Lee et al. (2020a), i.e., the **MUJOCO PUSH** task, is a planar pushing task, in which a 7-DoF Panda Franka robot is pushing a circular puck with its end-effector in simulation. We estimate the 2D position of the unknown object on a table surface while the robot intermittently interacts with the object. This dataset contains 1000 training data, 10 validation data, and 100 testing data, where each data point is split into 29 sequences, and each sequence includes 16 consecutive steps.

**V&T** Lee et al. (2020b), also called 'VISION&TOUCH', is a real-world robot manipulation dataset that collects visual, force, and robot proprioception data for a peg insertion task. The robot is used to insert the peg into the hole. In this paper, we use this dataset to predict the manipulator whether contacts with the peg in the next step, which is a binary classification task. We follow the setting of MultiBench and use $117,600$ data points for training and the remaining $29,400$ data points for validation and testing.

**ENRICO** Leiva et al. (2020) includes 20 Android app design categories. Each data point consists of the app screenshot and the view hierarchy. The view hierarchy describes the spatial and structural layout of UI elements of the corresponding screenshot. During training, the view hierarchy is ren-

Table 11: Concatenate tokens along the batch axis.

| Model | ENRICO ↑ | PUSH ↓ | AV-MNIST ↑ | Δ(%) ↑ |
|---|---|---|---|---|
| HighMMT multitask | 53.10 | 0.600 | 68.48 | 0.00 |
| SM$^4$ | **71.58** | **0.475** | **71.86** | **20.19** |
| Concate along batch | 64.38 | 1.174 | 71.05 | −23.57 |

dered as "wireframe", which can be viewed as a form of set data. ENRICO contains 947 data points for training, 219 data points for validation, and 292 data points for testing.

**AV-MNIST** Vielzeuf et al. (2018) is a multimedia dataset that uses audio and image information to predict the digit into one of 10 classes (0-9). This dataset comprises $55,000$ training data points, $5,000$ validation data points, and $10,000$ testing data points.

**UR-FUNNY** is the multi-modal affective computing dataset of humor detection in human speech. Each data point of UR-FUNNY is a video with text, visual, and acoustic modalities. We train this dataset to predict whether the current data point makes people feel positive or negative. There are $1,166$, $300$, and $400$ videos in the train, valid, and test data, respectively.

**MOSEI** Zadeh et al. (2018) is the largest dataset of sentence-level sentiment analysis and emotion recognition in real-world online videos. Each video is annotated for 9 discrete emotions (angry, excited, fearful, sad, surprised, frustrated, happy, disappointed, and neutral) and a continuous emotion value (valence, arousal, and dominance). We follow the MultiBench, training this dataset as a binary classification task. We use $16,265$, $1,869$, and $4,643$ train, valid, and test data points, respectively.

**MIMIC** Johnson et al. (2016), i.e., the Medical Information Mart for Intensive Care III, is a freely accessible critical care database, which records ICU patient data, including time-series and other demographic variables in the form of tabular numerical data. We use this dataset for binary classification on whether the patient fits any ICD-9 code in group 7 (460-519). The dataset is randomly split into $28,970$, $3,621$, and $3,621$ data points for training, validation, and testing.

For more details of the above datasets, please refer to the Liang et al. (2021) and their released website:

https://github.com/pliang279/MultiBench.

Results of HighMMT is running by Liang et al. (2022) released code:

https://github.com/pliang279/HighMMT.

### C.2 FUSION BY CONCATENATE TOKENS ON THE SEQUENCE DIMENSION

Before we input tokens into our transformer backbone (several consecutive transformer encoder layers), we concatenate tokens on the sequence dimension. Therefore, we can fuse different modalities by the attention layer within each transformer encoder layer. To further illustrate that such an operation is necessary, we additionally train the same model but concatenate tokens along the batch axis. Our following table shows fuse modalities by concatenating tokens along the sequence axis is positive for our tasks.

Our results in Table 11 show fuse modalities by concatenating tokens along the sequence axis is positive for our tasks.

### C.3 INDEPENDENT ROUTING POLICY BETWEEN Q, K, AND V

Prior works Fedus et al. (2022); Zhu et al. (2022) also apply MoE in the attention layer. However, they all use a single router to route tokens for q, k, and v simultaneously. We think such a design lacks flexibility. Therefore, in our MoE attention layer, the router for q, k, and v is separate, which could provide a more flexible attention mechanism. In order to support the above statement, we conduct additional experiments in Table 12 to study the advantage of SM$^4$ v.s. Prior MoE attention design style (q, k, v using the same router in the MoE attention).

Table 12: Using a single router to routing tokens for q, k, and v simultaneously.

| Model | ENRICO ↑ | PUSH ↓ | AV-MNIST ↑ | $\Delta$(%) ↑ |
|---|---|---|---|---|
| HighMMT multitask | 53.10 | 0.600 | 68.48 | 0.00 |
| SM$^4$ | **71.58** | **0.475** | **71.86** | **20.19** |
| qkv share routers | 73.51 | 0.936 | 69.28 | −5.45 |

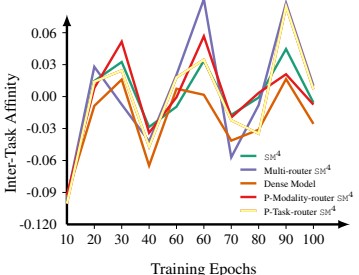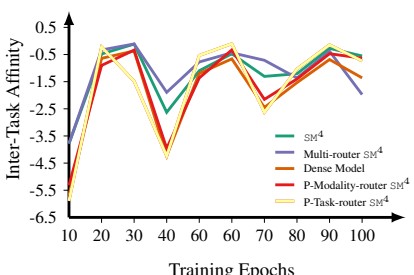

Figure 8: The inter-task affinity of the 'ENRICO' to the 'AV-MNIST' task ($left$), and the inter-task affinity of the 'ENRICO' to the 'PUSH' task ($right$). The results reported are the average of three replicates.

## C.4 THE GRADIENT POSITIVE SIGN PURITY OF SM$^4$

The Gradient Positive Sign Purity Chen et al. (2020) $\mathcal{P}$ of a single parameter for $T$ tasks is defined as:

$$\mathcal{P} = \frac{1}{2}(1 + \frac{\sum_i^T \Delta L_i}{\sum_i^T |\Delta L_i|}), \tag{5}$$

where $\Delta L_i$ is the gradient for the task $i$. The Gradient Positive Sign Purity is bounded by $[0, 1]$, which $\mathcal{P}$ close to 1 or 0 indicates such parameters suffer less gradient confliction from multi-task training. We use the trained model to collect the Gradient Positive Sign Purity of such a model. Then, we discrete the Gradient Positive Sign Purity value into five intervals of each parameter and count the ratio of parameters in these five intervals.

## C.5 THE TASK AFFINITY OF SM$^4$

The task affinity Fifty et al. (2021) is defined as follows:

$$\mathcal{Z}_{i \to j}^t = 1 - \frac{L_j(\mathcal{X}^t, \theta_{s|i}^{t+1}, \theta_j^t)}{L_j(\mathcal{X}^t, \theta_s^t, \theta_j^t)}, \tag{6}$$

where $\mathcal{X}^t$ is the training batch at time-step $t$, $\theta_{s|i}^{t+1}$ is the updated shared parameters after a gradient step with respect to the task $i$. $\theta_j^t$ represents the task $j$'s specific parameters. For the medium setting, we collect the task affinity by solitary training the 'PUSH' task for a single epoch, and then we calculate the loss of 'ENRICO' and 'AV-MNIST' on the corresponding training data. We count the task affinity from 'PUSH' to 'ENRICO' and 'AV-MNIST' every 10 epoch during training. We display the task affinity changes with training epochs in Figure 8. The task affinity of SM$^4$ and multi-router SM$^4$ is usually higher than the one of the dense model, which indicates that the MoE we proposed alleviates the training conflict of M$^3$TL.

## C.6 THE OPTIMAL GRADIENT BLEND OF SM$^4$

The optimal gradient blend Wang et al. (2020a) is used to re-weight the feature of each modality during multi-modal training. The optimal gradient blend will give this modality a small weight for the modality that is easy to prone to overfitting. The weight of each modality is bounded by $[0, 1]$ within a task, and the sum of all modalities for this task is 1. Therefore, the gap between different modalities within a task indicates that the modality with a smaller weight (optimal gradient

Table 13: The optimal gradient blend for each task under different model architectures.

| Model | ENRICO | | PUSH | | | | AV-MNIST | |
|---|---|---|---|---|---|---|---|---|
| | image | set | image | force | proprioception | control | image | audio |
| SM$^4$ | 0.48 | 0.52 | 0.00 | 0.37 | 0.32 | 0.31 | 1.00 | 0.00 |
| - Dense Model (w/o MoE) | 0.61 | 0.39 | 0.00 | 0.36 | 0.32 | 0.31 | 1.00 | 0.00 |
| - w/o Self-attention MoE | 0.63 | 0.37 | 0.00 | 0.37 | 0.32 | 0.30 | 1.00 | 0.00 |
| - w/o FFN MoE | 0.75 | 0.25 | 0.00 | 0.37 | 0.32 | 0.32 | 1.00 | 0.00 |
| multi-router SM$^4$ | 0.71 | 0.29 | 0.00 | 0.35 | 0.32 | 0.32 | 1.00 | 0.00 |
| P-Modality-router SM$^4$ | 0.73 | 0.27 | 0.00 | 0.37 | 0.31 | 0.32 | 1.00 | 0.00 |
| P-Task-router SM$^4$ | 0.80 | 0.20 | 0.00 | 0.36 | 0.32 | 0.31 | 1.00 | 0.00 |

blend) tends to overfit. We collect the optimal gradient blend of the corresponding trained model to determine whether our proposed model can restrain the easy model from overfitting. We use a modified version of the optimal gradient blend where the unnormalized optimal gradient blend of modality $m$ is defined as:

$$w_{unnorm}^{m,n} = \frac{L_{valid}^m}{L_{valid}^m - L_{train}^m}, \tag{7}$$

where $L_{valid}^m$ is the validation loss after training $n$ epochs only using modality $m$, and $L_{train}^m$ is the training loss after training $n$ epochs only using modality $m$. For task $i$, the final optimal gradient blend we reported is:

$$w_{i,m} = \frac{w_{unnorm}^{m,n}}{\sum_m^M w_{unnorm}^{m,n}}, \tag{8}$$

where $M$ is the number of modalities of the task $i$.

For M$^3$TL, the appropriate combination between modality-specific routers and task-specific routers (multi-router SM$^4$) helps each other better than purely using one of them (In Figure 8 and Table 13, the Inter-Task Affinity and the optimal gradient blend of multi-router SM$^4$ is better than models which only use modality-specific routers (P-Modality-router SM$^4$) or task-specific routers (P-Task-router SM$^4$)).

## C.7 Expert Selection Visualization

This section explores how tokens are distributed across different tasks and modalities by the routing policy of the SM$^4$. We show the expert selection of each routing policy under the testing distribution in Figure 9, Figure 10, and Figure 11. In these three settings, our routers work well, and most experts handle all modalities and tasks. Meanwhile, several experts focus on specific tasks.

For the large setting, we find out that the routing policy tends to route tokens to several specific experts, which also successfully proves MTL's MoE separate gradient conflict parameters. Especially for the 'MIMIC' dataset, only 2 to 4 experts are activated for this task.

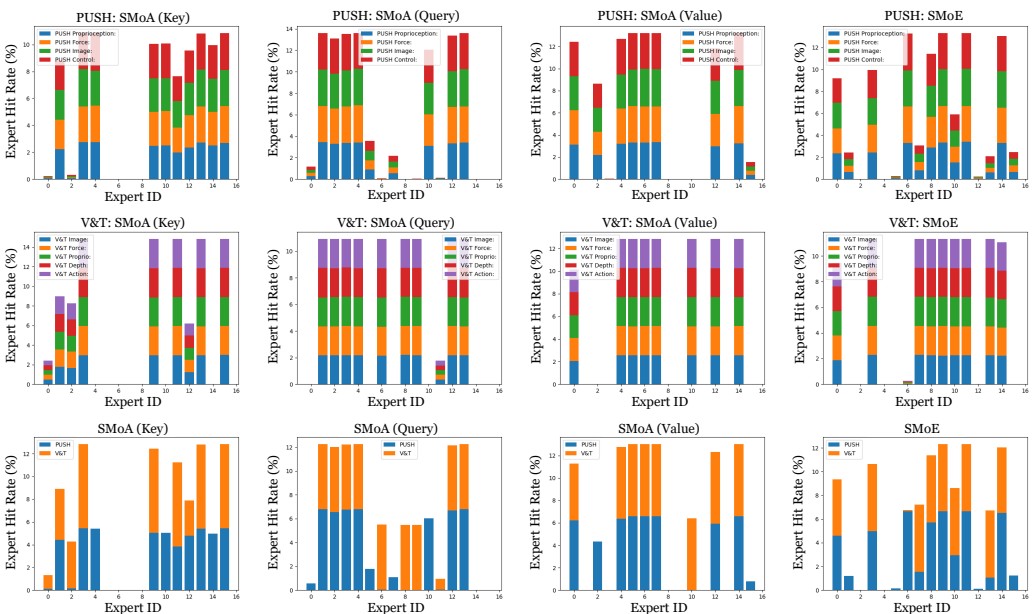

Figure 9: The expert selection of the small setting of the last SM$^4$ layer. The first two rows show the token distribution of different modalities for the 'PUSH' dataset and the 'V&T' dataset. The last row shows the token distribution across different tasks within three types of SMoA and SMoE.

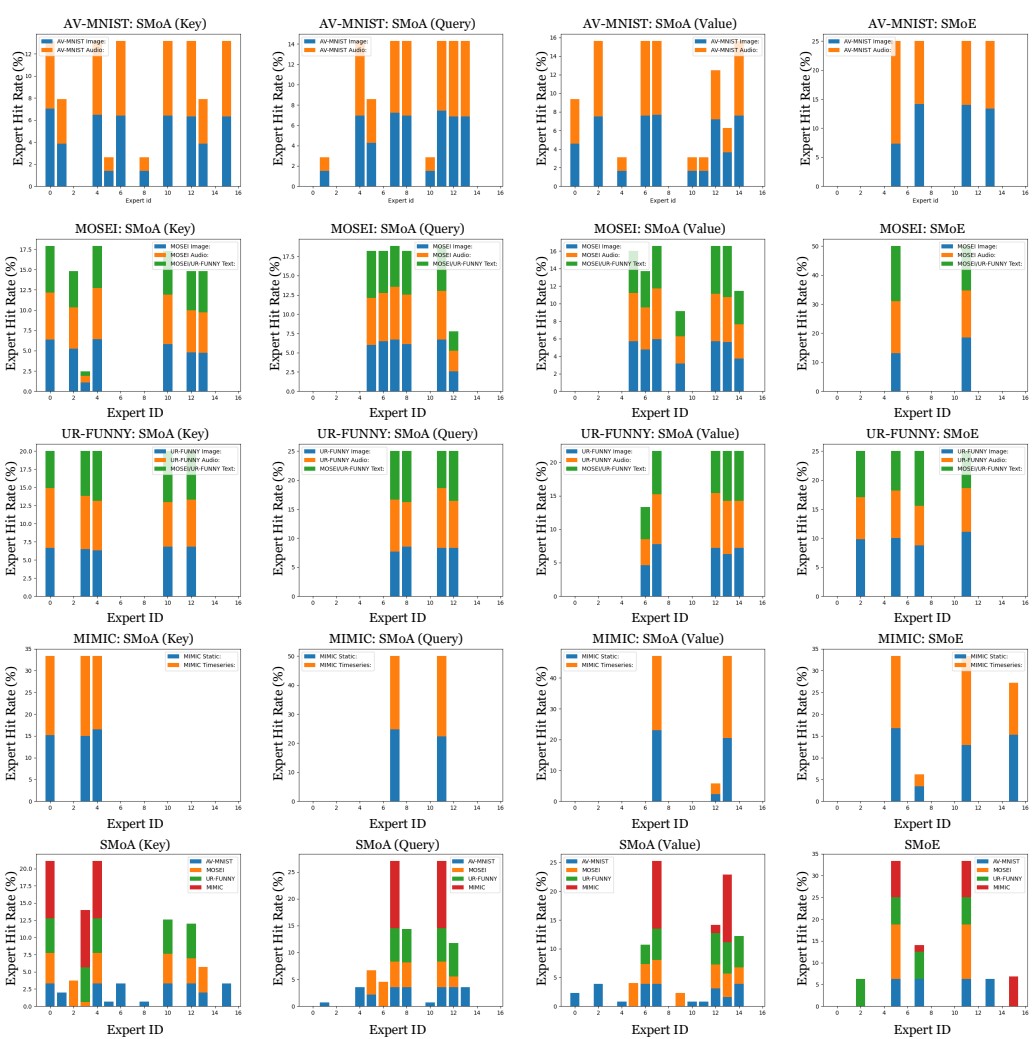

Figure 10: The expert selection of the medium setting of the last $\text{SM}^4$ layer. The first three rows show the token distribution of different modalities for the 'ENRICO' dataset, the 'AV-MNIST' dataset, and the 'PUSH' dataset. The last row shows the token distribution across different tasks within three types of SMoA and SMoE.

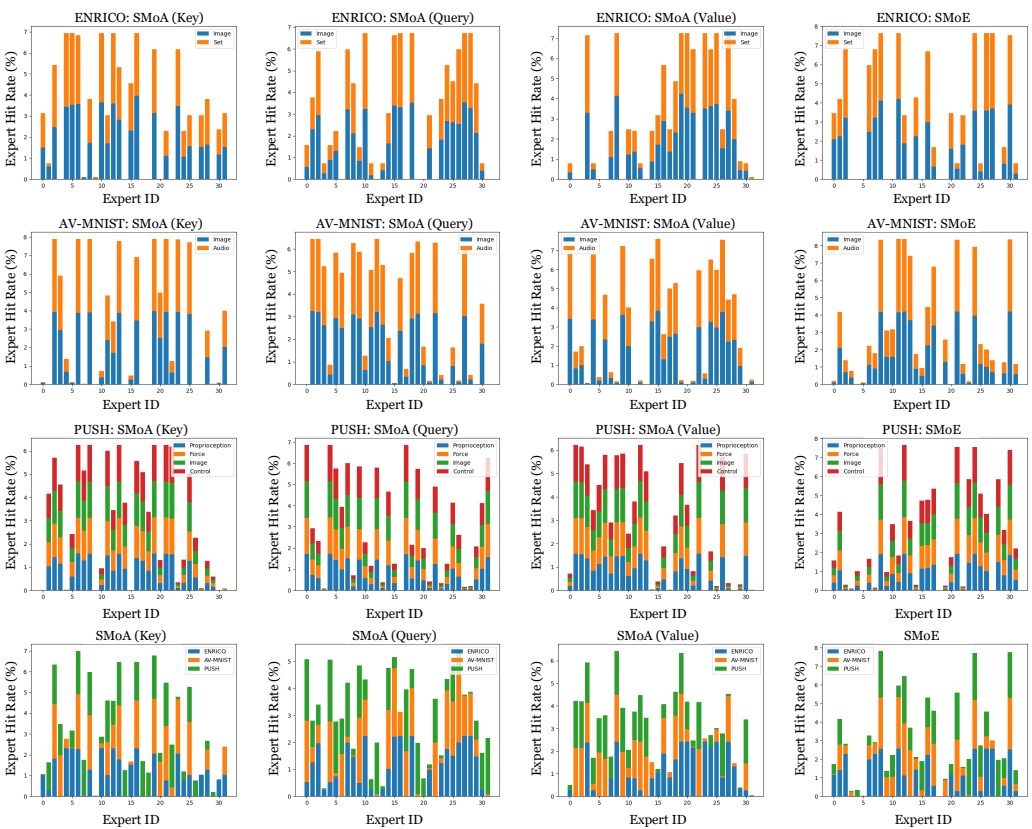

Figure 11: The expert selection of the large setting of the last $\text{SM}^4$ layer. The first four rows show the token distribution of different modalities for the 'AV-MNIST' dataset, the 'MOSEI' dataset, the 'UR-FUNNY' dataset, and the 'MIMIC' dataset. The last row shows the token distribution across different tasks within three types of SMoA and SMoE.

Table 14: Table of the modal and training setups on the small setting tasks: PUSH and V&T.

| Model Setup | | | |
|---|---|---|---|
| | Name of Hyperparameter | Value | |
| | | PUSH | V&T |
| Perceiver Unimodal Encoder | Sequence Length of Latent | 20 | |
| | Latent Dimension | 64 | |
| | Cross Attention Head | 1 | |
| | Cross Head Dim | 64 | |
| | Self-Attention Head | 8 | |
| | Self Head Dim | 64 | |
| MoE&MoA&Dense Encoder Layer | Depth | 1 | |
| | Self-Attention Head | 8 | |
| | Self Head Dim | 8 | |
| | Experts Number | 16 | |
| Classification Heads BatchNorm follow a Linear layer | Input/Output dimensions | 256/32 | 320/1 |
| Training | Optimizer | Adam | |
| | Learning rate | 0.0005 | |
| | Learning Scheduler | N/A | |
| | Weight Decay | 0.0 | |
| | Load&Importance Balancing Loss Weight | 0.1 | |
| | Pretrain | N/A | |
| | Max Epoch | 100 | |
| | Training loss weight | 100.0 | 1.0 |
| | Evaluation weight | 100.0 | 1.0 |
| | Batchsize | 28 | 64 |
| | Loss Function | MSE | CrossEntropy |
| MultiBench Input Dimension | | Gripper Pos: 16×3
Gripper Sensors: 16 × 7
Image: 16 × 32 × 32
Control: 16 × 7 | Image: 128 × 128 × 3
Force: 6 × 32
Proprio: 8
Depth: 128 × 128
Action: 4 |
| Dataset | Perceiver Input Channel Size | Gripper Pos: 3
Gripper Sensors: 7
Image: 1
Control: 7 | Image: 3
Force: 32
Proprio: 8
Depth: 1
Action: 4 |
| | Perceiver Input Extra Axis | Gripper Pos: 1
Gripper Sensors: 1
Image: 3
Control: 1 | Image: 2
Force: 1
Proprio: 1
Depth: 2
Action: 1 |
| | Perceiver Input num_freq_bands | Gripper Pos: 6
Gripper Sensors: 6
Image: 6
Control: 6 | Image: 6
Force: 6
Proprio: 6
Depth: 6
Action: 6 |
| | Perceiver Input max_freq | Gripper Pos: 1
Gripper Sensors: 1
Image: 1
Control: 16×7 | Image: 1
Force: 1
Proprio: 1
Depth: 1
Action: 1 |

Table 15: Table of the modal and training setups on the medium setting tasks: ENRICO, PUSH and AV-MNIST.

| Model Setup | | | | |
|---|---|---|---|---|
| | Name of Hyperparameter | Value | | |
| | | ENRICO | PUSH | AV-MNIST |
| Perceiver Unimodal Encoder | Sequence Length of Latent | 12 | | |
| | Latent Dimension | 64 | | |
| | Cross Attention Head | 1 | | |
| | Cross Head Dim | 64 | | |
| | Self-Attention Head | 8 | | |
| | Self Head Dim | 64 | | |
| MoE&MoA&Dense Encoder Layer | Depth | 1 | | |
| | Self-Attention Head | 8 | | |
| | Self Head Dim | 8 | | |
| | Experts Number | 32 | | |
| Classification Heads BatchNorm follow a Linear layer | Input/Output dimensions | 128/20 | 256/32 | 128/10 |
| Training | Optimizer | Adam | | |
| | Learning rate | 0.001 | | |
| | Learning Scheduler | CosineAnnealingLR | | |
| | Weight Decay | 0.0 | | |
| | Load&Importance Balancing Loss Weight | 0.05 | | |
| | Pretrain | Training PUSH for 100 epochs first | | |
| | Max Epoch | 100 | | |
| | Training loss weight | 10.0 | 10.0 | 0.8 |
| | Evaluation weight | 1.0 | 10.0 | 1.0 |
| | Batchsize | 32 | 32 | 32 |
| | Loss Function | CrossEntropy | MSE | CrossEntropy |
| MultiBench Input Dimension | | Image: $256 \times 128 \times 3$ Set: $256 \times 128 \times 3$ | Gripper Pos: $16 \times 3$ Gripper Sensors: $16 \times 7$ Image: $16 \times 32 \times 32$ Control: $16 \times 7$ | Colorless Image: $28 \times 28$ Audio Spectogram: $112 \times 112$ |
| Dataset | Perceiver Input Channel Size | Image: 384 (cut into $16 \times 8$ rectangles) Set: 384 (cut into $16 \times 8$ rectangles) | Gripper Pos: 3 Gripper Sensors: 7 Image: 16 (cut into $4 \times 4$ squares) Control: 7 | Colorless Image: 16 (cut into $4 \times 4$ squares) Audio Spectogram: 256 (cut into $16 \times 16$ squares) |
| | Perceiver Input Extra Axis | Image: 2 Set: 2 | Gripper Pos: 1 Gripper Sensors: 1 Image: 2 Control: 1 | Colorless Image: 2 Audio Spectogram: 2 |
| | Perceiver Input num_freq_bands | Image: 6 Set: 6 | Gripper Pos 6: Gripper Sensors: 6 Image: 6 Control: 6 | Colorless Image: 6 Audio Spectogram: 6 |
| | Perceiver Input max_freq | Image: 1 Set: 1 | Gripper Pos: 1 Gripper Sensors: 1 Image: 1 Control: 1 | Colorless Image: 1 Audio Spectogram: 1 |

Table 16: Table of the modal and training setups on the large setting include tasks: UR-FUNNY, MOSEI, MIMIC, and AV-MNIST.

| | Model Setup | | | | |
|---|---|---|---|---|---|
| | Name of Hyperparameter | Value | | | |
| | | UR-FUNNY | MOSEI | MIMIC | AV-MNIST |
| Perceiver Unimodal Encoder | Sequence Length of Latent | 12 | | | |
| | Latent Dimension | 64 | | | |
| | Cross Attention Head | 1 | | | |
| | Cross Head Dim | 64 | | | |
| | Self-Attention Head | 8 | | | |
| | Self Head Dim | 64 | | | |
| MoE&MoA&Dense Encoder Layer | Depth | 1 | | | |
| | Self-Attention Head | 8 | | | |
| | Self Head Dim | 8 | | | |
| | Experts Number | 16 | | | |
| Classification Heads BatchNorm follow a Linear layer | Input/Output dimensions | 192/2 | 192/2 | 128/2 | 128/10 |
| Training | Optimizer | Adam | | | |
| | Learning rate | 0.0008 | | | |
| | Learning Scheduler | N/A | | | |
| | Weight Decay | 0.001 | | | |
| | Load&Importance Balancing Loss Weight | 0.1 | | | |
| | Pretrain | N/A | | | |
| | Max Epoch | 100 | | | |
| | Training loss weight | 0.2 | 1.0 | 1.2 | 0.9 |
| | Evaluation weight | 1.0 | 1.0 | 1.0 | 1.0 |
| | Batchsize | 32 | 32 | 20 | 40 |
| | Loss Function | CrossEntropy | CrossEntropy | CrossEntropy | CrossEntropy |
| MultiBench Input Dimension | | Image: $20 \times 371$ Audio: $20 \times 81$ Text: $50 \times 300$ | Image: $50 \times 35$ Audio: $50 \times 74$ Text: $50 \times 300$ | Static: 5 Time-series: $24 \times 12$ | Colorless Image: $28 \times 28$ Audio Spectogram: $112 \times 112$ |
| Dataset | Perceiver Input Channel Size | Image: 371 Audio: 81 Text: 300 | Image: 35 Audio: 74 Text: 300 | Static: 1 Time-series: 12 | Colorless Image: 16 (cut into $4 \times 4$ squares) Audio Spectogram: 256 (cut into $16 \times 16$ squares) |
| | Perceiver Input Extra Axis | Image: 1 Audio: 1 Text: 1 | Image: 1 Audio: 1 Text: 1 | Static: 1 Time-series: 1 | Colorless Image: 2 Audio Spectogram: 2 |
| | Perceiver Input num_freq_bands | Image: 3 Audio: 3 Text: 3 | Image: 3 Audio: 3 Text: 3 | Static: 6 Time-series: 3 | Colorless Image: 6 Audio Spectogram: 6 |
| | Perceiver Input max_freq | Image: 1 Audio: 1 Text: 1 | Image: 1 Audio: 1 Text: 1 | Static: 1 Time-series: 1 | Colorless Image: 1 Audio Spectogram: 1 |

