# OpenReview forum: "Sparse MoE as a New Treatment: Addressing Forgetting, Fitting, Learning Issues in Multi-Modal Multi-Task Learning"
_ICLR.cc/2024/Conference — Submitted to ICLR 2024_

### Official Review · Reviewer_rXKX · 2023-10-27

**Soundness:** 2 fair
**Presentation:** 3 good
**Contribution:** 2 fair
**Rating:** 3
**Confidence:** 3

**Summary:**

This work tackles multi-modal multi-task learning with sparse mixture-of-experts. The authors identify three problems, namely modality forgetting, modality fitting and heterogeneous learning pace. The proposed method combines solutions for the three problems and shows competitive empirical performance against the SoTA.

**Strengths:**

This work identifies three important questions in multi-modal multi-task learning, namely forgetting, fitting and learning. Furthermore, the work proposes a framework that can solve the three problems simultaneously.

**Weaknesses:**

1. The novelty of the work is limited. To solve the modality forgetting problem, the authors deploy load and importance balancing loss. To solve the other two problems, the authors use standard hyperparameter tuning methods. It is unclear which part is truly originated from the authors.

2. The connection of the three problems is not organic. Although those three questions indeed exist in multi-modal multi-task learning, the authors do not point out how those problems are related. It seems that the authors tackle those three problems separately and in turn get a better result.

**Questions:**

I am confused about the results in 5. What does 32N mean? Intuitively, increasing N should lead to good performance, and 32N is indeed the largest in the table, so it is not surprising that it has the best result. What is the message to convey here?

---

> ### Author Response · Authors · 2023-11-22
> **Response to rXKX**
>
> ## **[Cons 1. The Novelty of The Work is Limited?]**
> We respectfully disagree. We point out its novelty richness from three distinct aspects:
> 1. [**SMoA Designs**] Our approach addresses the modality forgetting problem through the novel SM$^4$ structure, comprising SMoE and SMoA modules governed by modality-specific routing policies. **Note that the load and importance balancing loss is not designed for and can not solve the modality forgetting problem.**
> 2. [**Automatic Expert Allocation**] We emphasize that AEA is NOT a standard hyperparameter tuning. While hyperparameters rely on human intervention, our approach automatically decides the expert number based on the model training dynamics. Therefore, in our case, the expert number is not a hyperparameter.
> 3. [**Automatic Learning Pace for Different Modality**] The ALP automates the learning pace of each modality in a dynamic fashion, which is also not the standard hyperparameter tuning technique.
> ## **[Cons 2. The Connection of The Three Problems is Not Organic?]**
> We respectfully argue that these facets are intrinsically linked. Our proposal seeks to tackle the optimization problem in multi-modal multi-task learning, wherein the issues of gradient direction, optimization step size, and model capacity represent a comprehensive view of this overarching challenge.
> Our proposed SM$^4$ structure, featuring SMoA and SMoE modules alongside tailored training techniques, serves as a unified solution to alleviate these intertwined optimization challenges of multi-modal multi-task learning.
> ## **[Cons 3. Confused About The Results in 5. What Does 32N Mean?]**
> For the clarification of “32N”, we've revised the notation to N=4, N=8, N=16, and N=32.
> The optimal number of experts in Mixture-of-Experts (MoEs) remains a subject of debate, with conflicting views on the ideal quantity. While several studies advocate that increased experts enhance performance, an excessive number may introduce redundancy or noise. Table A3 of [1] demonstrates that increasing the expert number brings extra performance will gradually diminish as the expert number goes up, and Table 11 in [2] shows the large expert number may decrease performance.
> Therefore, we investigate this further for multi-modal multi-task learning, and our findings suggest that larger expert pools consistently yield performance improvements.
>
> [1] Mod-Squad: Designing Mixture of Experts As Modular Multi-Task Learners
>
> [2] TUTEL: Adaptive Mixture-of-Experts at Scaler

---

> > ### Author Response · Authors · 2023-11-23
> > **Response to rXKX**
> >
> > Dear Reviewer rXKX,
> >
> > We extend our heartfelt gratitude for your devoted time and meticulous review of our submission, and for the invaluable feedback you provided. Each of your comments has been meticulously assessed and duly addressed in our responses.
> >
> > Respectfully, we seek your evaluation of our rebuttal. Your assessment of how well we've addressed your concerns holds immense value for us. Additionally, we warmly welcome any supplementary remarks or discussions you may wish to engage in.
> >
> > Our sincere appreciation for your invaluable contributions and thoughtful consideration.
> >
> > Best regards,
> >
> > The Authors

---

> > > ### Comment · Reviewer_rXKX · 2023-12-03
> > >
> > > Dear authors,
> > >
> > > Thank you for the response, and Cons3 addresses my question.
> > >
> > > I am not convinced by the argument about the intrinsic connections between the three problems. Although those problems all fall under multi-modal multi-task learning, it does not mean that they necessarily need to be studied together unless there exists conceptual or empirical underpinning linking those problems. For instance, such link could be that solving one problem can help solve the other; or not solving one problem hurts the performance of the other. In this work, it is more like proposing three seperate solutions to each of the three questions. Due to the stack of three solutions to three problems, we would of course expect performance improvement compared with other algorithms only dealing one specific problem. Thus, I'll keep my score.

---

### Official Review · Reviewer_cbhp · 2023-10-27

**Soundness:** 3 good
**Presentation:** 3 good
**Contribution:** 2 fair
**Rating:** 5
**Confidence:** 4

**Summary:**

This work proposes SM$^4$ for the Multi-Modal Multitask Learning problem. Particularly, SM$^4$ focuses on the challenges of i) modality forgetting; ii) modality fitting; and iii) heterogeneous learning pace. SM$^4$ introduces several advances to the vanilla Sparse Mixture of Experts (SMoE) techniques, including employing SMoE in both the dense and multi-head self attention layers, implementing the adaptive expert allocation and adaptive learning pace mechanisms. SM$^4$ show promising results compared to SOTA baselines on the MultiBench benchmarks. Authors also conducted various ablation studies to explore different characteristics of SM$^4$.

**Strengths:**

- Multi-modal Multi-task learning is an important emerging problem in both research and industry.
- The proposed method  achieved encouraging performance against SOTA baselines.
- The experiments are quite comprehensive where the complexities and ablation studies are included. There are some exceptions that I will mention in the Weakness section.
- Implementation is available.

**Weaknesses:**

* My most critical concern of this work is the proposed method is quite ad-hoc and heuristic, especially in the AEA and ALP modules.
    + **AEA**: First, the strategy introduces an additional hyper-parameter: $n$ - number of iterations to monitor the loss. It is unclear how sensitive the results will be with respect to $n$, and there are no guideline to select $n$. Looking at Algorithm 2, it seems like AEA employs a pre-training phase to decide $k_j$ for each modality independently. However, this does not take into account the interaction of multitask learning when the modalities are learned together. There are also no constraints to enforce that all experts are utilized, i.e. $\sum_j k_j = N$. Lastly, in Figure 2-3, it is unclear why larger training-validation loss gap can lead to better generalization. When this gap is large, the model is either underfitted or overfitted rather than achieved better generalization.
    + **ALP**: it is unclear what "learning pace" mean in this context, i.e. is it the learning rate or some components that directly influence the training trajectory?

* Table 1 and 2 are quite unclear, what is the "setting" here referred to, is it the dataset size, or the model size? For example, in Table 2, SM$^4$ Medium - AV-MNIST has 1.23M params while the same method in the large setting has 0.76M params. The results of HighMMT seems to be quite different from the original, which requires further investigations.

* Other suggestions: Figure 2 is not nice, please consider using subfigure, e.g. 2a, 2b, etc. instead of the current presentation.

**Questions:**

- Clarifications regarding the AEA, ALP modules, and the settings in Table 1 & 2.

---

> ### Author Response · Authors · 2023-11-22
> **Response to cbhp**
>
> ## **Summary**
>
> Thank you for acknowledging the comprehensive of our experiments. Below are our responses addressing your concerns:
>
> ## **[Cons 1. AEA Introduces an Additional Hyper-Parameter, Additional Pre-training Phase?]**
> The AEA is not working on a pre-train phase; it is executed during our multi-modal multi-task learning, which takes into account multi-task learning, and the AEA stopped tuning expert numbers by itself, not a hyper-parameter.
> The “n” in Algorithm 2 is actually the number of iterations within a single multi-modal multi-task training epoch. Therefore, we do not include additional hyper-parameters.
> As we mentioned in Section 3.3, the load and importance balancing loss we used in routing networks make sure most experts will be utilized.
> Moreover, as shown in **T.1** our method doesn't increase training time significantly.
> The above concerns have been included in our revision.
>
> **T.1** SM$^4$ and HighMMT multi-modal multi-task training time.
> | Model   | Small Setting (h) | Medium Setting (h) | Large Setting (h) |
> | ------- | ----------------- | ------------------ | ----------------- |
> | SM$^4$  | 33                | 18                 | 8                 |
> | HighMMT | 33                | 17.5               | 8                 |
>
> ## **[Cons 2. Why Larger Training-Validation Loss Gap Can Lead to Better Generalization?]**
> As mentioned in [1], the ``generalization gap’’ is defined as the difference between a model’s performance on training data and its performance on unseen data drawn from the same distribution (e.g., the performance difference between training set and validation set). We adopt loss value to measure model performance and define the gap as the difference between training and validation loss. A larger generalization gap indicates that the model is not overfitted on the training set (i.e., higher training loss) but predicts well on the unseen data (i.e., lower validation loss), which indicates better generalization.
> We've revised the definition of the "generalization gap" to align with formal definitions, clarifying the significance of a higher training-valid loss gap indicating better generalization performance.
> ## **[Cons 3. What does "learning pace" Mean in This Context in ALP? ]**
> The learning pace in ALP refers to the learning rate (or the optimization step size) of different modalities.
> ## **[Cons 3. What is The "setting" in Table 1, 2 Refers to?]**
> The “setting” here refers to the number of tasks that follow the HighMMT.
> ## **[Cons 4. Model Parameter Number of Medium Setting is More Than The Large Setting.]**
> Although the large setting contains more tasks, more task number does not mean more parameters. As shown in Table 14 and Table 15 of our paper, the total number of experts in the medium setting is $32$, and we use $16$ experts in the large setting. Therefore, the number of model parameters in the medium setting is larger than in the large setting.
> ## **[Cons 5. The Results of HighMMT Seem to be Quite Different From The Origine.]**
> Our implementation of HighMMT adheres to their official repo [HighMMT] (https://github.com/pliang279/HighMMT) and the default settings outlined in the HighMMT paper. For instance, our utilization of learning rates is as follows: 0.0005 for the small setting, 0.001 for the medium setting, and 0.0008 for the large setting.
> Furthermore, our supplementary materials contain the HighMMT official code and reproduction scripts, ensuring the exact replication of all hyper-parameters specified in the HighMMT paper.
> If Reviewer cbhp could kindly point out another better re-implementation of HighMMT, we would like to follow it and update the configurations.
> ## **[Cons 6. Expression Suggestion.]**
> We've made adjustments according to your invaluable suggestions, including improvements to Figure 2 for clarity.
>
> [1] Predicting the Generalization Gap in Deep Networks with Margin Distributions.

---

> > ### Author Response · Authors · 2023-11-23
> > **Response to Reviewer cbhp**
> >
> > Dear Reviewer cbhp,
> >
> > We are grateful and appreciate your time and effort in reviewing our submission. We have thoroughly evaluated your comments and responded to them accordingly.
> >
> > As we near the end of the author-reviewer discussion, we kindly seek your feedback on our rebuttal. We would greatly appreciate your review of our responses and let us know if we have adequately addressed your concerns. Furthermore, we wholeheartedly welcome any additional comments or discussions.
> >
> > Thank you for your valuable input and consideration.
> >
> > Best regards,
> >
> > The Authors

---

> > > ### Comment · Reviewer_cbhp · 2023-11-23
> > > **Additional Clarification**
> > >
> > > Dear Authors
> > >
> > > Thank you for the detailed feedback. May I seek your clarifications on the following items.
> > >
> > > - [Cons 1. AEA and Algorithm 2] It is still unclear to me how Algorithm 2 is executed. In Algorithm 2, there is a step to **train the model for 1 epoch** (comment at line loss_val_i = train(model)). So is it that Algorithm 2 outlines the training procedure for the whole system? Can the authors provide a rough pseudo-code to outline how $SM^4$ is trained, and where AEA and ALP is performed at which step?
> > >
> > > - [Cons 4. Small - Medium - Large Settings] It is unclear to me the naming convention used here. From Table 1,  the PUSH dataset used in both small and medium setting, some datasets in the small or medium settings are even larger than those in the large settings (e.g. V&T in small). From your response just now and in Table 2, the small models can have more parameters and more FLOPS than those in medium or large. So what is the size of the setting refer to in this experiment?

---

> > > > ### Author Response · Authors · 2023-11-23
> > > > **Response to Reviewer cbhp**
> > > >
> > > > Thank you for your feedback, we present specific responses below:
> > > >
> > > > ## **[Cons 1. AEA and Algorithm 2]**
> > > > > Does Algorithm 2 outline the training procedure for the whole system?
> > > >
> > > > No, Algorithm 2 does not outline the training procedure for the whole system. It only describes the behavior of Automatic Expert Allocation during multi-modal multi-task training.
> > > >
> > > > > Rough pseudo-code of SM$^4$
> > > >
> > > > The rough pseudo-code of SM$^4$ is provided below.
> > > > - After each epoch, we will collect the loss and the routing entropy information of each modality and then utilize this information to tune the topk expert number and the learning pace of each modality accordingly.
> > > > - Notably, in the Automatic Expert Allocation (AEA), if the specific modality is signed as **"improved"**, AEA will skip the action of tuning the topk value of this modality.
> > > > ```python
> > > > def overall_training(modality_topk, modality_weights):
> > > > 	for i in range(max_epochs):
> > > > 		# training 1 epoch
> > > > 		# losses include the average loss of all modalities in this epoch
> > > > 		# routing_entropy includes average entropy of all modalities in this epoch
> > > > 		losses, routing_entropy = train(model, modality_topk, modality_weights)
> > > >
> > > > 		# setting topk of all modalities
> > > > 		# if the monitoring is ended, AEA will skip to tuning the topk of specific modality
> > > > 		modality_topk = AEA(model, losses)
> > > >
> > > > 		modality_weights = ALP(model, routing_entropy)
> > > > ```
> > > > ## **[Cons 4. Small - Medium - Large Settings]**
> > > > > What is the size of the setting referred to in this experiment?
> > > >
> > > > The setting referred to **the number of tasks** in this experiment, where the small setting includes two tasks, the medium setting includes three tasks, and the large setting includes four tasks.
> > > >
> > > > Again, heartfelt gratitude for your devoted time and invaluable comments.

---

> ### Comment · Reviewer_cbhp · 2023-11-23
> **Training model step in AEA**
>
> Dear authors,
>
> In Algorithm 2, there is a step that state $loss\\_val\\_i = train(model)$ with a comment **training model for 1 epoch**. So does this step reuse the losses in the pseudo-code that you outlined or it requires re-training the model for 1 epoch?
>
> If we directly plug Algorithm 2 in the overall_training pseudo-code, I guess there might be a lot of nested training loops in the early stage because each AEA step also requires measuring the validation loss of each modality.
>
> Algorithm 2 also instructs to continue to train the model for the remaining epochs (2nd last line). So I think it needs to be revised to integrate to the overall_training pseudo-code.
>
> Lastly, this is a conceptual questions. Since the authors stated that the expert allocation to a modality is kept until end of training when the flag improved is set to true. This suggests that early stages of training will mostly perform a search to allocate experts to modalities and this configuration will remain fixed. Is it an optimal strategy? Because in early epochs, the experts may not learn well enough and maybe it is beneficial to "reallocate" the experts once they have learned good representations. One naive strategy could be perform a reallocation (re-run AEA) after some fixed epochs.

---

> ### Author Response · Authors · 2023-11-23
> **Response to Reviewer cbhp about the AEA**
>
> > Does this step reuse the losses "loss_val_i" in the pseudo-code?
>
> Yes, this step reuses the losses in the pseudo-code.
>
> > Revise the Algorithm 2
> Thanks to the suggestion, we revised the Algorithm for integrating the overall_training pseudo-code and modified overall_training pseudo-code below:
> ```
> def adaptive_expert_allocation(model, loss):
>     for modality in modality_set:
> 	    # If the modality is signed improved, skip this modality
> 	    if check(modality):
> 		    continue
> 		n_experts = modality_topk[modality]
>         loss_val = loss[valid][modality]
> 		# if the expert number of this modality is increased last time
> 		if increase_expert(modality):
> 			if loss_decrease(loss_val):
> 				improved = True
> 		else:
> 			# if the valid loss does not decrease
> 			if loss_decrease(loss_val):
> 				if not improved:
> 					Sign this modality as improved
> 					modality_topk[modality] = n_experts - 1
> 				else:
> 					modality_topk[modality] = n_experts + 1
> 					improved = False
>     return modality_topk
> ```
> ```python
> def overall_training(modality_topk, modality_weights):
> 	for i in range(max_epochs):
> 		# training 1 epoch
> 		# losses include the average valid loss of all modalities in this epoch
> 		# routing_entropy includes the average entropy of all modalities in this epoch
> 		val_losses, routing_entropy = train(model, modality_topk, modality_weights)
>
> 		# setting topk of all modalities
> 		# if the monitoring of specific modality is ended, AEA will skip to tuning the topk of specific modality
> 		modality_topk = AEA(model, val_losses)
>
> 		modality_weights = ALP(model, routing_entropy)
> ```
>
> > Is it an optimal strategy?
>
> This problem needs further investigation.
> Reallocation is a potential solution to further improve the performance of SM$^4$. The problem is this will involve additional hyper-parameters to decide when to reallocate experts which makes the training more complex. However, we think this is a remarkably interesting and important problem, how to decide the optimal model capacity of each modality during training, and expect to investigate this further in our future work.
>
> Thanks for your insightful comments!

---

> > ### Comment · Reviewer_cbhp · 2023-12-04
> > **Acknowledgement**
> >
> > I appreciate the authors for engaging in the discussion. After careful considerations, I believe the current manuscript will be tremendously benefitted from another revision to clarify the training procedure and some experimental protocols. Although my view towards this work became more positive, I can't recommend for acceptance at the current state.

---

### Official Review · Reviewer_mdyv · 2023-10-31

**Soundness:** 3 good
**Presentation:** 4 excellent
**Contribution:** 3 good
**Rating:** 8
**Confidence:** 3

**Summary:**

The authors propose a novel framework, SM^4, based on sparse Mixture-of-Exports and designed for multi-modal multi-task learning. Notably, the algorithms address three 3 critical issues in the field: modality forgetting, overfitting to simple modalities, and unaligned learning paces in multi-tasks. The main idea is to disentangle information and adjust model capacities by enforcing sparsity and employing attention models. In the experiments, SM^4 shows the best performance, greatly reduced computational cost, and the ability of mitigating the 3 pain points.

**Strengths:**

1. This work is well-motivated and addresses important issues in multi-modal and multi-task learning via reasonable algorithms. The evaluations and analyses are also detailed, confirming the impact of this work.
2. The conducted analyses not only prove the effectiveness of the framework SM^4 but also establish solid evaluation protocol for follow-up works.
3. The writing is impressive. The authors do a great job on presenting the complicated settings and methods, making the article both informative and easy to follow. Also, the experiment settings are thoroughly reported.

**Weaknesses:**

My concerns are mostly about the experiments.
1. The authors employ MultiBench for evaluation, while the metrics of robustness and training cost are ignored. This raises 2 concerns:
* a. Without checking robustness, it is unclear if the trained model is robust to missing or noisy modalities, which shall be an important criterion in multimodal learning.
* b. The trade-off between training cost and model performance of SM^4 is unclear. In particular, deciding number of experts for each modality seems to be time-consuming. I suppose checking the trade-off and comparing SM^4 with simple methods such as early/late can help measuring the practical value of this work more precisely.
2. The reported performance of MultiBench models in Table 2 may be overly simplified. As the complexities of the MultiBench models greatly vary, simply reporting the aggregated performance (e.g., the range) makes it difficult to position SM^4 in this regard. Also, the dependencies between efficiency and performance are ignored, similar to the issue in weakness 1.b. A candidate method could be the 2D visualization adopted by MultiBench and is used for studying trade-offs.
3. Minor typo: YR-FUNNY in Table 1.

**Questions:**

1. Following the weaknesses, I am wondering if the authors consider reporting the training cost, robustness, and the trade-offs?

---

> ### Author Response · Authors · 2023-11-22
> **Response to mdyv 1/3**
>
> ## **Summary**
>
> We are glad that reviewer mdyv appreciates the comments on our writing as “impressive”, “informative”, and “easy to follow”, our proposal as “well-motivated and addresses important issues”, and our experiments as “detailed” and “solid”. To address reviewer mdyv’s questions, we provide pointwise responses below.
>
> ## **[Cons 1. Without Checking Robustness.]**
>
> We conduct extra experiments to examine the robustness of SM$^4$ to missing modalities. Due to the limited time in the rebuttal period, we will add the robustness evaluation of noisy modalities in our final version.
> As shown in Table **T1**, we assess SM$^4$ and HighMMT with the model UR-FUNNY under three missing scenarios of missing text, video, and audio, respectively. Our results imply that SM$^4$ has a relatively better robustness towards missing modalities.
>
> **T.1** The robustness comparison between SM$^4$ and highMMT. We show the $\delta$ value, which is defined as the value of performance drop when missing one modality. A smaller $\delta$ value indicates better robustness against the modality missing.
>
> | UR-FUNNY| Missing text | Missing video | Missing audio |
> | -------------------| ------------ | ------------- | ------------- |
> | SM$^4$ | $1.16$ | $4.77$ | $0.92$ |
> |HighMMT| 8.22 | 10.30 | 6.62 |
>
> | MOSEI| Missing image | Missing audio| Missing text |
> | -------------------| ------------ | ------------- | ------------- |
> | SM$^4$ | $0.39$| $0.82$ | $0.92$|
> |HighMMT| 10.45 | 17.38 | 12.23 |
>
> | MIMIC | Missing table| Missing timeseries|
> | -------------------| ------------ | ------------- |
> | SM$^4$ | $8.71$| 17.99 |
> |HighMMT| 9.72 | $11.67$ |
>
> | AV-MNIST| Missing image| Missing audio|
> | -------------------| ------------ | ------------- |
> | SM$^4$ | 60.6| $13.60$ |
> |HighMMT| $55.85$ | 36.27 |
>
> ## **[Cons 2. The Trade-off Between Training Cost and Model Performance of SM^4 is Unclear.]**
>
> Thanks for the great point.
> - The total number of training epochs for SM$^4$ and other baselines are the same. The procedure of deciding the number of experts for each modality does NOT require extra training epochs. Specifically, both HighMMT and SM$^4$ use $100, 100, 100$ training epochs for the small, medium, and large settings, respectively.
> - To further convince review mdyv, we measure the training time of SM$^4$ and HighMMT in the same device (i.e., single NVIDIA A30 GPU). Our results demonstrate that SM$^4$ uses **similar training costs** to reach consistently enhanced performance, as evidenced in Table **T.2**. It indicates an improved trade-off between training cost and model performance.
>
> **T.2** SM$^4$ and HighMMT multi-modal multi-task training time.
> | Model   | Small Setting (h) | Medium Setting (h) | Large Setting (h) |
> | ------- | ----------------- | ------------------ | ----------------- |
> | SM$^4$  | 33                | 18                 | 8                 |
> | HighMMT | 33                | 17.5               | 8                 |
>
>
> ## **[Cons 3. The Reported Performance of MultiBench Models in Table 2 Maybe Overly Simplified.]**
> To address the concern regarding the detailed performance of MultiBench models, we included a more comprehensive performance report below (Table **T.3**, **T.4**, **T.5**, and **T.6** below) and in our revision (Table 10 (page 22) in our revision).
>
> Also, the typo in Table 1 has been corrected in our revision.

---

> ### Author Response · Authors · 2023-11-22
> **Response to mdyv 2/3**
>
> **T3**. Detailed performance, parameter usage, and FLOPS of multi-modal learning method on MultiBench benchmark. For the ''FLOPS(G)'', ''-'' indicates the MultiBench does not provide official implementation. Notably, the empty FLOPS of the MultiBench Model (MFAS)'' is due to the FLOPS of ''MFAS'' being dynamic during training. For each dataset, we choose multi-modal models with the best/worst performance and multi-modal models with the largest/smallest parameter numbers, respectively.
> | Method                             | Dataset           | Performance | \# Parameter (M) | FLOPS (G) |
> | ---------------------------------- | ----------------- | ----------- | ---------------- | --------- |
> | MultiBench Models (TF-LSTM)        | PUSH $\downarrow$ | 0.574       | 23.5             | 25.11     |
> | MultiBench Models (LF-LSTM)        | PUSH $\downarrow$ | $0.290$       | $1.90$             | $14.07$     |
> | MultiBench Models (MULT)             | PUSH $\downarrow$ | 0.402       | 14.6             | 19.20     |
> | MultiBench Models (LRTF)             | V\&T              | 93.3        | $1.09 $            | $5.20$      |
> | MultiBench Models (LF)             | V\&T              | $93.6$        | 1.20             | $5.20$      |
> | MultiBench Models (RefNet)         | V\&T              | 93.5        | 135              | $-$       |
> | MultiBench Models (TF)             | ENRICO            | 46.6        | $19.3$             | 314.13    |
> | MultiBench Models (GradBlend)      | ENRICO            | $51.0$        | $19.3$             | 314.13    |
> | MultiBench Models (RefNet)         | ENRICO            | 44.4        | 25.7             | $2.67$      |
> | MultiBench Models (GradBlend)      | AV-MNIST          | 68.5        | 0.29             | 0.50      |
> | MultiBench Models (MFAS)           | AV-MNIST          | $72.8$        | $0.14$             | $-$       |
> | MultiBench Models (RefNet)         | AV-MNIST          | 70.9        | 14.1             | $0.25$      |
> | MultiBench Models (EF-GRU)         | UR-FUNNY          | 60.2        | 3.58             | 3.13      |
> | MultiBench Models (MULT)           | UR-FUNNY          | $66.7$        | 2.38             | 3.37      |
> | MultiBench Models (MCTN)           | UR-FUNNY          | 63.2        | $0.19$             | $0.17$      |
> | MultiBench Models (TF)             | UR-FUNNY          | 61.2        | 12.2             | 2.67      |
> | MultiBench Models (MCTN)           | MOSEI             | 76.4        | $0.19$             | $0.15$      |
> | MultiBench Models (MULT)           | MOSEI             | $82.1$        | 4.75             | 3.35      |
> | MultiBench Models (LF-Transformer) | MOSEI             | 80.6        | 31.5             | 21.6      |
> | MultiBench Models (MI-Matrix)      | MIMIC             | 67.9        | 0.801            | $0.005 $    |
> | MultiBench Models (LF)             | MIMIC             | $68.9$       | 0.034            | $0.005$     |
> | MultiBench Models (LRTF)           | MIMIC             | 68.5        | $0.008$            | $0.005$          |
>
> **T4**. Performance, parameter usage, and FLOP of our model, and HighMMT in the small setting.
> | Method  | Dataset           | Performance | \# Parameter (M) | FLOPS (G) |
> | ------- | ----------------- | ----------- | ---------------- | --------- |
> | HighMMT | PUSH $\downarrow$ | 0.445       | 0.89             | 5.14      |
> | HighMMT | V\&T              | 96.10       | 0.85             | 32.48     |
> | SM$^4$  | PUSH $\downarrow$ | $0.331$       |$0.27 $            | $2.59$      |
> | SM$^4$  | V\&T              | $96.33$       | $0.25$             | $17.38$          |
>
> **T5**. Performance, parameter usage, and FLOP of our model, and HighMMT in the medium setting.
> | Method  | Dataset           | Performance | \# Parameter (M) | FLOPS (G) |
> | ------- | ----------------- | ----------- | ---------------- | --------- |
> | HighMMT | ENRICO            | 53.10       | $0.58 $            | 79.48     |
> | HighMMT | PUSH $\downarrow$ | 0.600       | $0.63$             | 21.60     |
> | HighMMT | AV-MNIST          | 68.48       | $0.52 $            | 0.95      |
> | SM$^4$  | ENRICO            | $71.58 $      | 1.23             | $1.10$      |
> | SM$^4$  | PUSH $\downarrow$ |$ 0.475$       | 1.25             | $2.33$      |
> | SM$^4$  | AV-MNIST          | $71.86$       | 1.23             | $0.41$          |

---

> ### Author Response · Authors · 2023-11-22
> **Response to mdyv 3/3**
>
> **T6**. Performance, parameter usage, and FLOP of our model, and HighMMT in the large setting.
> | Method  | Dataset  | Performance | \# Parameter (M) | FLOPS (G) |
> | ------- | -------- | ----------- | ---------------- | --------- |
> | HighMMT | UR-FUNNY | 62.00       | $0.52$             | 1.51      |
> | HighMMT | MOSEI    | 78.40       | $0.52$             | 1.65      |
> | HighMMT | MIMIC    | 65.60       | $0.52$             | 0.67      |
> | HighMMT | AV-MNIST | 70.60       | $0.52$             | 0.95      |
> | SM$^4$  | UR-FUNNY | $64.24$       | 0.76             | $0.38$      |
> | SM$^4$  | MOSEI    | $79.47$       | 0.76             | $0.53$      |
> | SM$^4$  | MIMIC    | $67.91$       | 0.76            | $0.15$      |
> | SM$^4$  | AV-MNIST | $71.05$       | 0.76             | $0.43$          |

---

> > ### Author Response · Authors · 2023-11-23
> > **Response to Reviewer mdyv**
> >
> > Dear Reviewer mdyv,
> >
> > We express our gratitude for your dedicated time and meticulous review of our submission, along with the insightful feedback provided. Every comment you offered has been thoroughly assessed and addressed in our responses.
> >
> > We respectfully seek your assessment of our rebuttal. Your evaluation of our responses regarding your concerns would be immensely valuable to us. Moreover, we invite any supplementary remarks or discussions you may wish to initiate.
> >
> > We extend our sincere thanks for your invaluable contributions and thoughtful consideration.
> >
> > Best regards,
> > The Authors

---

> > > ### Comment · Reviewer_mdyv · 2023-11-23
> > >
> > > I appreciate the authors' detailed responses and agree the responses address my concerns. I have adjusted the score accordingly. Nevertheless, I would like to echo the concern from reviewer m8oX and cbhp regarding the performance gap of HighMMT between Liang et al., 2022 and this work. I have read the authors' response about this issue, while I believe some discussion or at least an acknowledgement needs to be included in this manuscript.

---

> > > > ### Author Response · Authors · 2023-11-23
> > > > **Thanks to Reviewer mdyv**
> > > >
> > > > Dear Reviewer mdyv,
> > > >
> > > > Thank you for your positive feedback. Your insights are invaluable in guiding our ongoing research efforts. Meanwhile, we agree that a discussion about the performance of HighMMT is needed, and we added it to our revision.
> > > >
> > > > Best regards,
> > > >
> > > > The authors.

---

### Official Review · Reviewer_m8oX · 2023-11-01

**Soundness:** 3 good
**Presentation:** 2 fair
**Contribution:** 3 good
**Rating:** 5
**Confidence:** 4

**Summary:**

The paper presents an approach that incorporates routing in the training of general-purpose models for multimodal and multi-task learning to address heterogeneity across modalities and tasks. The overall idea is relatively straightforward. A subset of datasets from MultiBench were used in their evaluation.

**Strengths:**

+ The performance of the proposed method on included datasets seems impressive based on numbers reported in the paper.
+ Solid study on the behavior of the proposed method. Plots in Figure 2 are good illustrations, right to the points.

**Weaknesses:**

- The description of proposed method is very difficult to follow. I have no idea how the method works from just reading the paper. For example, it does not clearly state what are exactly the experts and where they come from. I cannot get much from Figure 1. It seems that experts are grouped. But I was not able to find a discussion why/how they are grouped.

- The motivation of ALP is not clear to me. Does unstable routing policy just mean changes in the policy across iterations? Such change does not necessarily link to the routing distribution entropy.

- Comparing Table 2 with Table 3 of Liang, et al., 2022, there is large difference in the performance of HighMMT on same datasets. A discussion of where the discrepancy coming from is needed.

**Questions:**

- What are the criteria/considerations used in selecting datasets/tasks for evaluation? There are many other tasks and modalities in MultiBench. How the proposed method works for those?

In addition, refer to the list of weaknesses

---

> ### Author Response · Authors · 2023-11-22
> **Response to m8oX**
>
> ## **[Summary]**
> Acknowledgments to Reviewer m8oX for recognizing our work as "impressive" and appreciating the clarity of our Figure 2, outlining its directness. We value the constructive feedback provided, which is instrumental in refining our paper. To address reviewer m8oX's queries, we present specific responses below:
>
> ## **[Cons 1. What are Exactly the Experts, and Where do They Come From? Are They Grouped? .]**
>
> The experts are duplicated multi-layer perceptrons (MLP). Specifically, in SMoE, the experts are duplicated from the original feedforward networks. The experts in SMoA are duplicated from the query, key, and value MLPs, respectfully.
>
> In order to distinguish experts from different sources, we name them as different expert groups. **Experts are grouped by the nature of where they are duplicated from.**
>
> The above clarifications have been included in our revisions.
>
> ## **[Cons 2. Motivation of ALP. How do the Policy Changes Link to the Routing Distribution Entropy?]**
>
> 1. *What does an unstable policy mean?*
>
> An unstable routing policy means the changes in the policy across iterations.
>
> 2. *How do such changes link to routing distribution entropy?*
>
> They are linked. We measure the routing distribution of each expert across the iterations and then calculate an averaged entropy across all the experts. Therefore, the entropy here directly measures the stability of the routing policy over the training iterations. For example, a high entropy indicates that expert routing is quite converged and stable.
>
> 3. *Motivation of ALP.*
>
> The motivation of adaptive learning pace (ALP) is to align different learning paces between modalities, aiming to synchronize the optimization of multiple objectives. To be specific, from Figure 2(d), the stability of modality-specific routing seems to reveal the convergence of modality-specific training, serving as a good guidance to decay its learning rate.
> The above clarifications have been included in our revisions.
>
> ## **[Cons 3. Performance Difference of HighMMT.]**
>
> Thanks for pointing it out.
> - We highlight that our performance of HighMMT is produced with the official implementation from HighMMT’s repository (https://github.com/pliang279/HighMMT). We strictly follow the default configurations reported in their paper, as shown in Tables 8, 9, and 10. For example, we use learning rates of 0.0005, 0.001, and 0.0008 for the small, medium, and large settings, respectively.
> - Additionally, our supplementary materials have included the HighMMT code and reproduction scripts, maintaining exact replication of their hyper-parameter settings.
> - If Reviewer m8oX could kindly point out another better re-implementation of HighMMT, we would like to follow it and update the configurations.
>
> ## **[Cons 4. Datasets/Tasks Selection.]**
>
> The selection of our three multi-modal task groups follows HighMMT’s standards.
> - The small setting encompasses similar research areas with varying modality inputs;
> - The medium setting spans three domains featuring different modalities;
> - The large setting involves three domains incorporating diverse modalities.
>
> Yes, our proposed method is a general multi-modal, multi-task pipeline, which can be leveraged for various combinations of tasks and modalities. **In our submission, we follow the standard multi-modal multi-task settings from the HighMMT paper**. That is, HighMMT/Multibench currently only provides the default hyper-parameter configurations for the small, medium, and large settings, which are adopted in our paper as the representative baseline. In our future works, we will examine our methods for extra combinations of tasks and modalities.

---

> > ### Author Response · Authors · 2023-11-23
> > **Thanks to Reviewer m8oX**
> >
> > Dear Reviewer m8oX,
> >
> > We extend our sincere gratitude for your invaluable time and commitment in evaluating our manuscript. Acknowledging the constraints on your schedule during this busy period, we respectfully seek your input on our rebuttal, as the discussion phase approaches its conclusion (the discussion period will end in less than 12 hours).
> >
> > If you have any additional comments or suggestions regarding our manuscript, we eagerly welcome the opportunity to engage in further discussion with you.
> >
> > Looking forward to your response.
> >
> > With deepest gratitude,
> >
> > The Authors

---

### Meta-Review · Area_Chair_k3As · 2023-12-15

**Metareview:**

This paper proposes a mixture of experts methd for multi-modal multi-task learning for tackling relevant problems of modality forgetting, task imbalance, and optimization challenges. The reviewers have raised concerns on the novelty and clarity of the solutions

(1) The authors attempt to address the modality forgetting problem through standard losses like load balancing and importance weighting.
The solutions to task imbalance and optimization issues rely more on extensive hyperparameter tuning rather than new techniques.

(2) the connection between the three problems is not clearly articulated. It seems they are solved in isolation without demonstrating interdependence. This reduces the coherence of the approach.

I would encourage the authors to consider a major revision, and the paper could be much stronger if these concerns could be properly addressed.

**Justification For Why Not Higher Score:**

NA

**Justification For Why Not Lower Score:**

NA

---

### Decision · Program_Chairs · 2024-01-16

Reject